# Controls over debris flow initiation in glacio-volcanic environments in the Southern Andes.

Ivo Fustos-Toribio[1], Daniel Basualto[3*], Ardy Gatica[1], Alvaro Bravo-Alarcón[2,1], José-Luis Palma[4], Gabriel Fuentealba[5,2], Sergio A. Sepúlveda[6,7]

[1]Departamento de Ingeniería en Obras Civiles, Facultad de Ingeniería y Ciencias, Universidad de La Frontera. Francisco Salazar #01145, Temuco, Chile.
[2] Programa de Magíster en Ciencias de La Ingeniería, Universidad de La Frontera, Temuco, Chile.
[3]Departamento de Ingeniería Eléctrica, Facultad de Ingeniería y Ciencias, Universidad de La Frontera. Francisco Salazar #01145, Temuco, Chile.
[4]Departamento Ciencias de la Tierra, Facultad de Ciencias Químicas, Universidad de Concepción, Víctor Lamas 1290, Concepción, Chile.
[5]Ministerio del Interior, Temuco, Chile.
[6] Departamento de Geología, Facultad de Ciencias Físicas y Matemáticas, Universidad de Chile, Santiago, Chile
[7] Department of Earth Sciences, Faculty of Science, Simon Fraser University, Burnaby, Canada

*Correspondence to*: daniel.basualto@ufrontera.cl

**Abstract.** The southern Andes is an active zone of mass wasting processes with unknown constraints for public policies. Several conditioning factors could have an impact on the generation of debris flows, being controlled by water accumulation. This study investigates the generation of the Ñisoleufu debris flow, an active area of debris flow generation in Southern Andes, reviewing the interplay between geomorphological, geotechnical and hydrometeorological controls in debris flow dynamics, focusing on the effects of soil properties, slope characteristics and precipitation events.

Our results highlight significant changes in soil moisture content on critical days associated with debris flow events. We revealed that the combination of areas with high water accumulation capacity from local runoff and slopes that capture precipitation effectively were crucial in the generation of debris flows. Areas with granular volcanic soils acted as storage mediums for water, which, coupled with decreased shear strength, facilitated debris flow initiation. The thin and fine-grained layers of glacial deposits located beneath the volcanic soil, characterized by low hydraulic conductivity, created localized accumulation zones that reinforced the storage capacity of adjacent areas, particularly in pyroclastic volcanic deposits in the release zone. The hydraulic properties of the volcanic deposits suggest that water storage capacity and high hydraulic conductivity play a critical role in rainfall-induced debris flow initiation. Additionally, we observed that the debris flow of the Ñisoleufu event has evidence of reworked lapilli-sized particles (>5 mm), being consistent with the surface and shallow water movement that reduces the slope stability within the area.

Analysis of ERA5-land dataset showed abrupt changes in soil moisture content at various depths and time periods, correlating with intense or prolonged rainfall events. These results underscore the role of geomorphological features in modulating soil

moisture and thereby affecting the stability and movement of debris flows. Our results provide a comprehensive understanding of how geomorphology interacts with hydrological factors to influence debris flow behaviour in volcanic areas of the Southern Andes for the first time. Overall, the research highlights the critical role of geomorphological and hydrological factors in debris flow generation and dynamics. It emphasizes the need for incorporating detailed soil and slope characteristics into models for predicting debris flow risks. By understanding the combined effects of water accumulation, soil properties, and slope dynamics, this study contributes valuable insights into managing and mitigating debris flow hazards in vulnerable regions. These findings enhance the predictive capacity for rainfall-induced debris flows and provide practical criteria for hazard assessment in post-glacial volcanic terrains.

## 1    Introduction

Episodes of extreme rainfall have increased due to climate change, resulting in a greater frequency of debris flows (Jakob and Lambert 2009; Lee 2017; Fustos et al., 2017; Dey and Sengupta 2018) and mainly related to fast changes of soil water content (Fustos et al., 2021). Currently, mass wasting processes produce widespread damages, representing a significant threat to human life (Sepúlveda and Petley 2015; Vega and Hidalgo 2016). Consequently, the need to forecast (Fustos et al., 2020a) and mitigate (Fustos et al., 2021b) the effects of these events has become a high priority for governments facing increasing episodes of rainfall-induced landslides (RIL) linked with climate change. An accurate assessment of potential debris flows to regional scale needs precise understanding of their triggering and controlling conditions. Therefore, we assessed the main conditioning and triggering conditions of debris flows in Southern Andes in order to understand their evolution from stable slope to mass wasting event.

Worldwide, the increasing frequency and intensity of such extreme precipitation events exacerbated by climate change to regional scale (Stoffel et al., 2013; Pavlova et al., 2018) introduce severe threats to human life and property. Changing precipitation patterns related to extreme precipitation, lead to conditions conducive for debris flows in wide areas in North America (Bovis et al., 1999), Asia (Chang et al., 2017), Europe (Malet et al., 2005; Stoffel et al., 2013; Pavlova et al., 2018) and South America (Sepúlveda et al., 2013; Sepúlveda et al., 2014; Fustos et al., 2022). Stand out heavy tropical storms, such as Taiwan, increasing debris flow incidents (Chang et al., 2025) and compromising the safety of urban areas located near mountainous terrains (Chen et al., 2015; Kang et al., 2017). The Wenchuan Earthquake in China exemplifies how seismic activity can trigger extensive debris flows, resulting in not only immediate destruction but also long-lasting hazards due to the formation of landslide-dammed lakes that threaten downstream communities (Cui et al., 2009; Wang and Yan 2015). Accurate forecasting of debris flow events is vital for disaster preparedness and mitigation. Understanding the triggering conditions—including rainfall intensity, groundwater levels, and geological features—is essential for developing conceptual models that represent the regional conditions that could control a debris flow initiation. Debris flows are influenced by soil hydraulic characteristics and the intensity/duration of rainfall events (Singh and Kumar, 2020), in which rainfall intensities serve as crucial predictors in mountainous regions (Chang et al., 2017; Fustos-Toribio et al., 2022). Moreover, coarse-grained volcanic soils exhibit transient increases in pore pressure during intense rainfall events (Huang et al., 2012). Conversely, fine-grained soils with low infiltration rates do not experience significant changes in the pore pressure, generating failures due to decreased soil shear strength (Dahal et al., 2008; Dahal et al., 2011). Hence, understanding soil composition and granulometric features is pivotal in assessing debris flow susceptibility worldwide. Debris flows are mainly controlled by the geomorphological features and the specific geological evolution of each region of the planet, highlighting the need for localized and context-specific approaches for their study and management. One of the next frontier corresponds to constrain the debris flow generation inglacial environments under changing precipitation events related to climate change.

Nowadays, understanding the impact of debris flows in glacial environments becomes critical in the Chilean southern Andes, particularly due to the most part of the inhabitants live there. Changes of precipitation patterns related to climate change,

particularly fast and intense rainfall events, could amplify the frequency and magnitude of debris flows (Fustos et al., 2022). An increase of extreme hydrometeorological events affecting slopes in glacial settings is observed, whose mechanical properties and geomorphology have evolved since the Last Glacial Maximum (Fustos-Toribio et al., 2021b; Somos-Valenzuela et al., 2020; Ochoa-Cornejo et al. 2024). Considerable uncertainty remains about how the interaction between volcanic-derived soils over glacial landforms will respond to extreme hydrometeorological events. One of the current models for initiation of debris flows is  related to slow deforming surfaces in hillslopes, mainly due to gravity and surface erosion during high precipitation events (Xie et al. 2020; Yi et al. 2021).  Slow surface deformation could lead to extensional failures that could expand and deepen, generating landslides and evolving into debris flows, especially under water-saturated conditions or heavy rainfall (Gregoretti, 2000; Fustos et al., 2017; Wang et al., 2024). The capacity to oversee these extensional failures in remote areas close to roads is an open question yet, mainly in Southern Andes.

Historical debris flow events in the Southern Andes remain subject to uncertain conditioning factors. The transformation of landslides into debris flows typically occurs when sliding material incorporates water, significantly increasing its fluidity—as observed in the Villa Santa Lucía event in Chilean Patagonia (Somos-Valenzuela et al., 2020). The occurrence of debris flows in volcanic settings is of considerable scientific interest (Cheung & Giardino, 2023; Sepúlveda et al., 2008), largely due to the complex nature of volcanic soils and their marked textural variability (Thompson et al., 2023), which strongly influence water retention and infiltration dynamics. Recurrent debris flows in the Osorno volcano exemplify the role of intense rainfall events and the mobilization of autobrecciated lava blocks in initiating such processes (Fustos et al., 2022). Although numerical modeling has estimated total flow volumes (ranging from $4.7\times10^5$ to $5.5\times10^5$ m³) highlighting the high sensitivity of debris flow generation to the initial water content, a comprehensive conceptual model remains lacking. Over the past four decades, the Southern Andes has experienced significant volcanic activity (Galetto et al., 2023), resulting in widespread tephra deposition that has contributed to increased frequencies of debris flows (Korup et al., 2019). Nevertheless, critical knowledge gaps persist regarding the spatial and temporal variability of the textural and hydraulic properties of volcanic soils, particularly under extreme hydrometeorological conditions. Addressing these gaps is crucial to better constrain the hydraulic and geomechanical conditions that lead to debris flow initiation (Schmidt  et al., 2001; Kuriakose et al., 2009). Such understanding is fundamental for effective territorial planning, risk mitigation, and the development of robust early warning systems.

In this paper, we analyse the precursory surface deformation related to tensile cracks leading to the Ñisoleufu debris flow event (May 31[st], 2021). We utilised a multi-temporal InSAR approach with Sentinel-1 C-band SAR data, enabling us to create time series plots of deformation before the landslide and compare them with available daily precipitation records from nearby weather stations and satellite measurements. By combining remote sensing data, weather records, and soil laboratory analyses, we aim to provide valuable insights into the factors leading to such events and, consequently, improve hazard assessment and management in zones with soils derived from explosive volcanic events.

## 2    Study area

Debris flows are the most common manifestation of mass wasting triggered by precipitation in the Southern Andes due to the soil heterogeneity and geological features, providing a unique opportunity to study the relationship between extreme rainfall and debris flows (Figure 1 A-D). Recent extreme precipitation events have produced mass wasting hazard, especially in steep zones near alluvial plains where human settlements are often established (Fustos et al., 2017; Fustos-Toribio et al., 2021). On May 31st, 2021, a very fast debris flow was triggered due to extreme rainfall affecting houses and blocking the CH-201 route in the Ñisoleufu zone, southern Chile, generating economic losses in a vulnerable rural area (Figure 1B; Figure 2). The deposit mainly consisted of rock blocks and trunks covered by a thin layer of debris. Much of the debris flow fell into the adjacent Calafquén lake (Figure 2), causing a small tsunami. The event was extremely rapid, based on the classification proposed by Hungr et al., (2014), with a speed estimated to be over 3 m/s. Stand out that the debris flow experienced reactivation events on June 19, 2023, and again on June 28, 2024.

The debris flow was deposited in a flat area of the valley (slopes between 0-20°), flowing into Lake Calafquén. The presence of landslide deposits and old flows in the vicinity of the lake (Figure 2A; geologic map [Ha]) suggests that this phenomenon is common in the area.

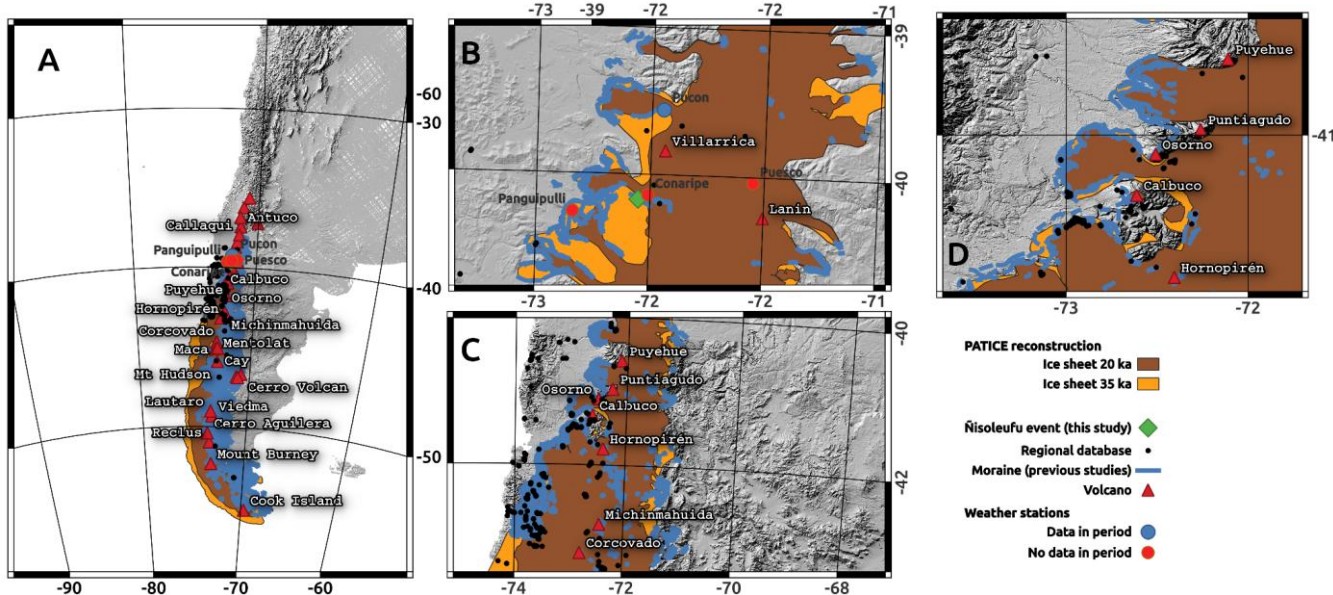

**Figure 1 Rainfall-Induced mass wasting in Southern Andes. A) Regional map of Ice-sheet extension during 35 ka and 20 ka as example and volcanoes emplaced in the area. B) Zoom to study area with Ñisoleufu in Northen Ice sheet sector showing the weather stations. C) Zoom to Northern Patagonian area showing correlation between mass wasting events and moraine lines (blue line). D) Zoom to Osorno volcano area showing high debris flow generation area discussed in Fustos et al., 2022.**

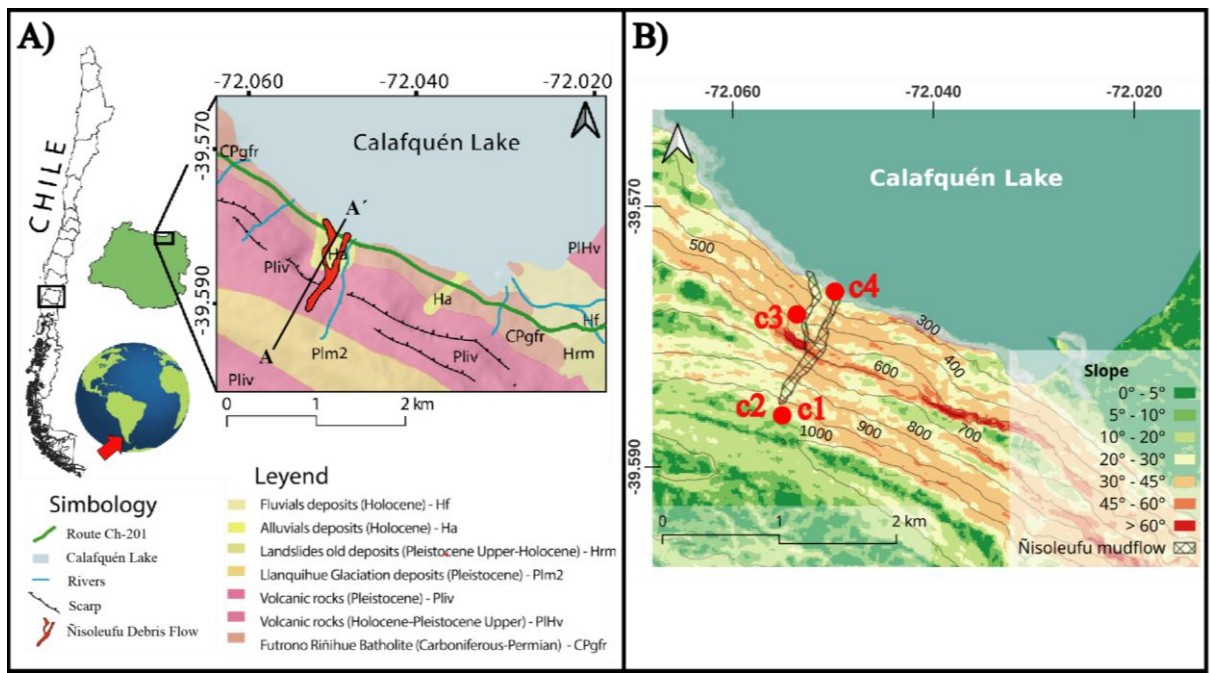

**Figure 2 A) geological map of area study based on Rodriguez et al. (1999). B) slope and elevation values (m.a.s.l.).**

The Ñisoleufu area exhibits a geological sequence **(**Figure 2A**)** starting at the base with the Futrono-Riñihue Batholith (CPgfr), followed by stratified volcano-sedimentary units (Pliv) and topped by glacial deposits (Plm2). Within this sequence, the Sierra Quinchilca volcano-sedimentary unit has been dated to less than 1 million years and is in nonconformity contact with the underlying rocks of the Futrono-Riñihue Batholith. Consolidated glacial deposits from the last glacial maximum period show variable depths, ranging from 1 metre to 10 centimetres, and lie in erosive unconformity over the volcano-sedimentary units (Pliv). The area also exhibits mass movement deposits (Hrm and Ha) covering the older units on the north slope of Sierra Quinchilca (Pliv, Figure 2A**;** Rodríguez et al., 1999).

## 3   Methodology

To assess the triggering and conditioning factors in glacial environments in Southern Andes, we assessed in detail the Ñisoleufu debris flow (Figure 3). To achieve this, we employed two complementary methodologies. The first methodology involved fieldwork, including soil sampling and subsequent laboratory analysis to evaluate the geotechnical features influencing debris flow initiation. The second methodology utilized numerical models to analyse meteorological conditions in the study area, such as precipitation patterns, variations in soil moisture, and landforms associated with erosion and deposition zones linked to the landslide. This endeavour involved a meticulous examination of the rheological and hydraulic attributes of the soil, facilitating an understanding of past and contemporary surface processes. Through this analytical framework, it became

feasible to discern causative factors contributing to the event, thereby enabling an informed evaluation of the potential risk

posed by analogous occurrences in the future.

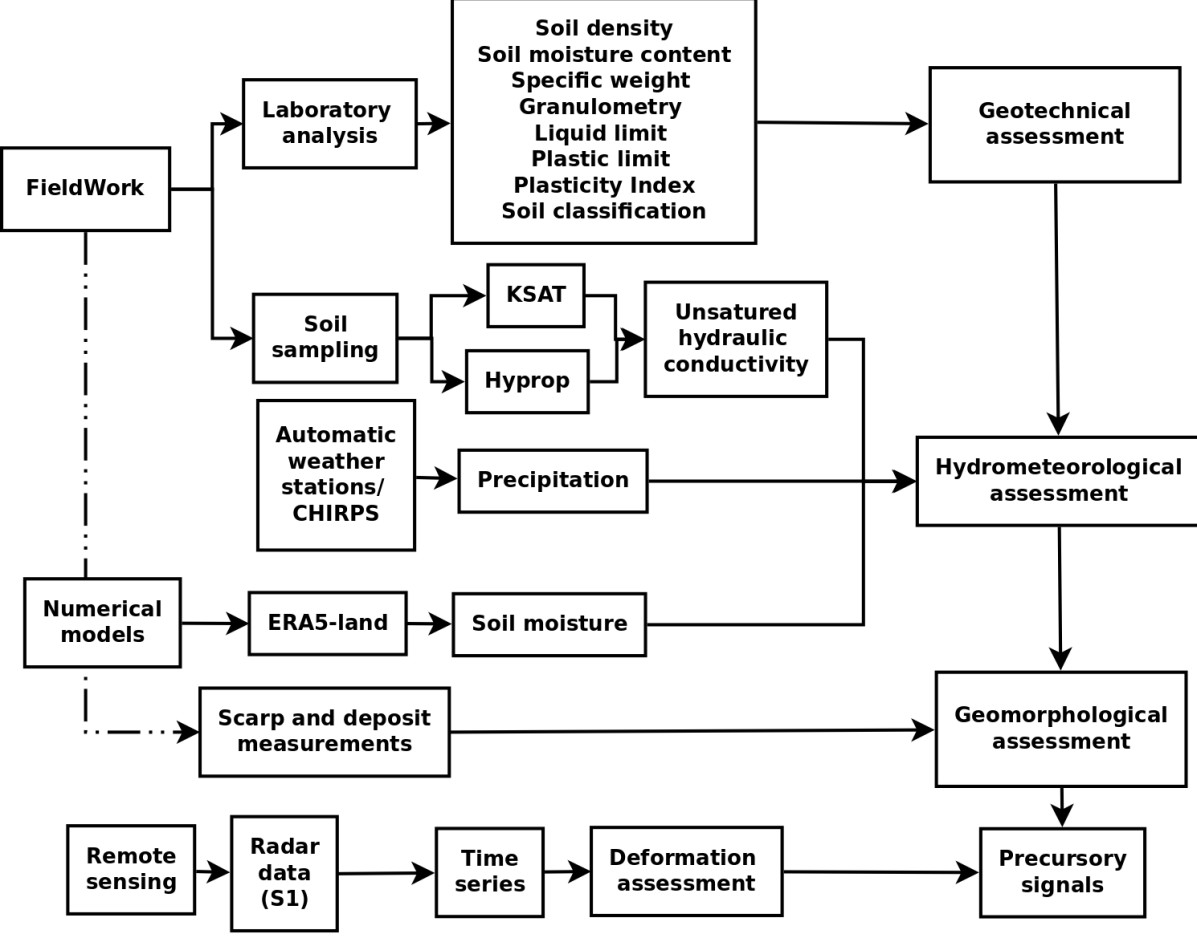

**Figure 3 Methodological approach.**

## 3.1    Geomorphological and geotechnical conditions

To analyse the geomorphological characteristics of the area, we utilised a 12.5 m resolution ALOS PALSAR DEM data, from

which slope, aspect, and elevation maps were derived. A terrestrial and remote sensing survey was carried out to characterise

the physiography of the hillslope before and after the debris flow.  We employed the Normalized Difference Vegetation Index

(NDVI) to delineate the area affected by the debris flow. We calculated the NDVI using two Sentinel-2 acquisitions on May

13th, 2021, and June 14th, 2021, corresponding to the Ñisoleufu debris-flow event on May 31st, 2021. Field campaigns were

conducted one day and three months after the event (June 1st, 2021 and September 2021) to characterise post-event

geomorphological features. Field observations were determinant in the identification of the sediment budget of the channel, aiding the assessment of the event's characteristics (a channelised, stony debris flow) and the movement dynamics in the hillslope. On-channel deposits were assessed using cross-sections along the Ñisoleufu sector. Peak flow marks were documented, and erosion depths were estimated based on erosion marks and bedrock exposures over three cross-sections.

Moreover, an exhaustive analysis of the geomorphological changes resulting from the event was conducted. A detailed survey of the stratigraphic column was conducted at three key points (Points c1, c2 and c4 in Figure 2B). Firstly, a stratigraphy sequence at the base of the debris flow was generated, obtaining the deposit sequence and assessing previous non-documented events. Secondly, a survey was conducted in the headscarp area where the event was initiated, evaluating the geological and geomorphological conditions that led to its generation. This approach allowed for the characterisation of the strata and structures in the affected area, seeking insights into the triggering mechanisms of the flow. Additionally, a detailed evaluation of the lateral erosion caused by the flow was carried out to understand the impact of local geomorphological changes as proxies for similar glacial-volcanic environments (Bucher et al., 2024). Finally, we assessed changes in vegetation that occurred as a result of the May 31st event and its subsequent reactivations (Figure 2B), allowing the estimation of the evolution of the landscape.

To analyse the geotechnical features that support a debris flow generation or another type of mass wasting antecedent, soil samples were collected in the generation zone. We determined soil density ($\rho$) following the UNE 103-301-94, soil moisture content (NCh 1515), specific weight (ASTM D854-14) standards. Granulometry analysis was carried out using sieve and hydrometer methods based on Kinde et al. (2024). The liquid limit was determined using AS 1289.3.9.1, and the plastic limit was evaluated following NCh 1517/2 standard, allowing to obtain the Plasticity Index. Finally, the soils were classified based on ASTM D2487-17. These comprehensive geotechnical analyses provided crucial insights into the soil properties and their potential role in the occurrence of the debris flow event.

### 3.2    Hydrometeorological conditions

To analyse the hydrometeorological conditions, we investigate the influence of rainfall in the study area, analysing hourly/daily data from four weather stations (Figure 1B) from the INIA agrometeorological and DMC networks (https://agrometeorologia.cl/) and the Climate Hazards Group InfraRed Precipitation with Station (CHIRPS) precipitation estimates (Funk et al., 2015). Soil moisture data from the ERA5-land product was utilised to complement the analysis considering the antecedent soil moisture at different depths before debris flow (Bordoni et al., 2023; Palazzolo et al., 2023), being considered suitable due to their accurate soil moisture data in hydrological cases (Muñoz-Sabater et al., 2021). The ERA5 product provides valuable and reliable information on soil moisture, enabling a more comprehensive understanding of the hydrological conditions that could trigger debris flows in the area. To understand the water transfer capacity along the soil, we measured the layer thicknesses from visual assessment of the soil profile in the scar (Figure 4). We followed the experimental design of De Pue et al., (2019) considering two samples per layer. One sample was used to measure the soil moisture ($\Theta$ (h)) and unsaturated hydraulic conductivity ($K_u$) using the evaporation method (HYPROP®, Meter Group),

meanwhile, saturated hydraulic conductivity $K_s$ was measured using KSAT® equipment (Meter Group), using the falling head method (Dane et al., 2002). The hydraulic conductivity will provide valuable information about the layer's ability to allow water to flow through, which could have played a significant role in the initiation and propagation of the debris flow.

**3.3 Remote sensing and precursory signals**

We assessed precursory signals estimating surface deformation by the Stanford Method for Persistent Scatterers (StaMPS; Hooper et al., 2008; Hooper et al., 2012) using Sentinel-1 C-band SAR data. We downloaded 35 ascending orbit (track 164) and 18 descending orbit (track 83) Sentinel-1 images covering the period from November 2020 to June 2021, which includes seven months before the May 31st debris flow in the Ñisoleufu sector (Table 1). We also included two acquisitions after the debris flow in both orbits to analyse the slope's response to non-rainfall and rainfall periods. Initially, we analysed data until the end of July, but snow coverage led to coherence loss in the area, resulting in a low density of Persistent Scatterer points (PS points) per km$^2$. Consequently, we evaluated different combinations of bands from Sentinel-2 images (based on bands 4, 3, 2) to assess the maximum amount of SAR images available before the snow period. This approach allowed us to optimise the data selection process and continue our analysis effectively.

The SAR data was processed using open-source Sentinel Application Platform (SNAP) packages through the snap2stamps routines, enabling us to generate single-master interferograms compatible with StaMPS. Further details on the snap2stamps routine in Foumelis et al. (2018) and Blasco et al. (2018). First, the initial selection of PS points is performed based on their noise characteristics, using the amplitude dispersion criterion, which is defined by $D_{Amp} = \sigma_{Amp}/m_{Amp}$, where $\sigma_{Amp}$ and $m_{Amp}$ are the standard deviation and mean of the amplitude in time, respectively (Ferretti et al., 2001). We selected a threshold value of 0.4 for $D_{Amp}$ as a typical threshold value (Hooper et al., 2007), and subsequently, some initial parameters were modified according to the values proposed by Höser (2018). This allowed us to plot surface soil deformation using time series, which we then compared with daily precipitation records obtained from nearby weather stations and satellite measurements. Lastly, we used the GACOS correction (Yu et al., 2018) through the TRAIN toolbox (Bekaert et al., 2015) to reduce the atmospherical phase component. We complemented surface deformation with antecedent precipitation to establish precursory signals and their correlation with precipitation. We calculated the accumulated precipitation between the dates of the SAR acquisitions and the total accumulated precipitation for the entire period. We assessed daily precipitation and their temporal changes to understand the antecedent precipitation in the debris flow event. The four weather stations were identified within a reasonable radius for potential use as sources of meteorological data. However, upon examining the temporal coverage and continuity of their records, it was found that only the Pucón station had complete and operational data for the study period. Moreover, we merged this data with CHIRPS dataset for comparison with the time series of deformation to examine the relationship between precipitation and deformation.

Table 1 Data used in temporal analysis using SAR data.

|     | Track | Frame | First Image | Last Image | Total Images | Primary Image | Sub-Swath |
| --- | --- | --- | --- | --- | --- | --- | --- |
| **DES** | 83 | 93 | 02 November 2020 | 18 June 2021 | 18 | 02 March 2021 | IW2 |
| **ASC** | 164 | 1048 | 01 November 2020 | 17 June 2021 | 35 | 11 February 2021 | IW3 |

## 4    Results

### 4.1    Geomorphological and geotechnical conditions

The extension of the area affected by the debris flow was evaluated, identifying and characterizing the triggering conditions (Figure 5A). The results revealed significant differences in the Normalized Difference Vegetation Index (NDVI), which facilitated the delimitation of the landslide area (Figure 5A). Low positive NDVI values (<0.20) are shown for the area affected by the debris flow, encompassing 118,575 m², in contrast to the normally high NDVI values (0.6-0.8) of surrounding areas. The debris flow moved along a complex and abrupt geometry, with slopes varying from 20° near the ridges (950 m. a.s.l.) and the base of the slope (250 m a.s.l.), to almost vertical areas (~90°) in the intermediate zone (550 m a.s.l.) (Figure 4A **and** Figure 5D). This geomorphological configuration is typical of glacially eroded valleys (U-shaped valleys) in the southern Andes, a recurrent phenomenon in the formation of valleys at this latitude (Muratli et al., 2010).

 The initial landslide crown has an altitude of 950 m. a.s.l. and slopes of approximately 20 to 30° (column c1 in Figure 4B). The flow release zone (inset C in Figure 4) has evidence of extensional failures, where c1 indicates that the first level S-1 corresponds to a very compact chaotic and polymictic till deposit (Plm2), with a greyish matrix containing a higher percentage of clay than sand. Some clasts exceed 10 cm and are composed of volcanic fragments (Pliv), as well as intrusive material (CPfgr). Towards the top of the glacial deposit, level S-2 is observed, a thin, grey fluvioglacial deposit (varves), approximately 40 cm thick, composed of a well-consolidated matrix of clay-rich (dark) and silt-rich (light) setting a couplet annual sediment layer. (Figure 4C, Figure 4D **and** Figure 5D). Above the glacial deposit, level S-4 is identified, composed mainly of lapilli-sized deposits (>5 mm), associated with pumice from the Neltume deposit of the Mocho-Choshuenco Volcanic Complex, dating 10,200+-500 BP (Rawson et al., 2015; Moreno-Yaeger et al., 2024). Finally, level S-7 corresponds to the current soil where native forest develops (Figure 4B).

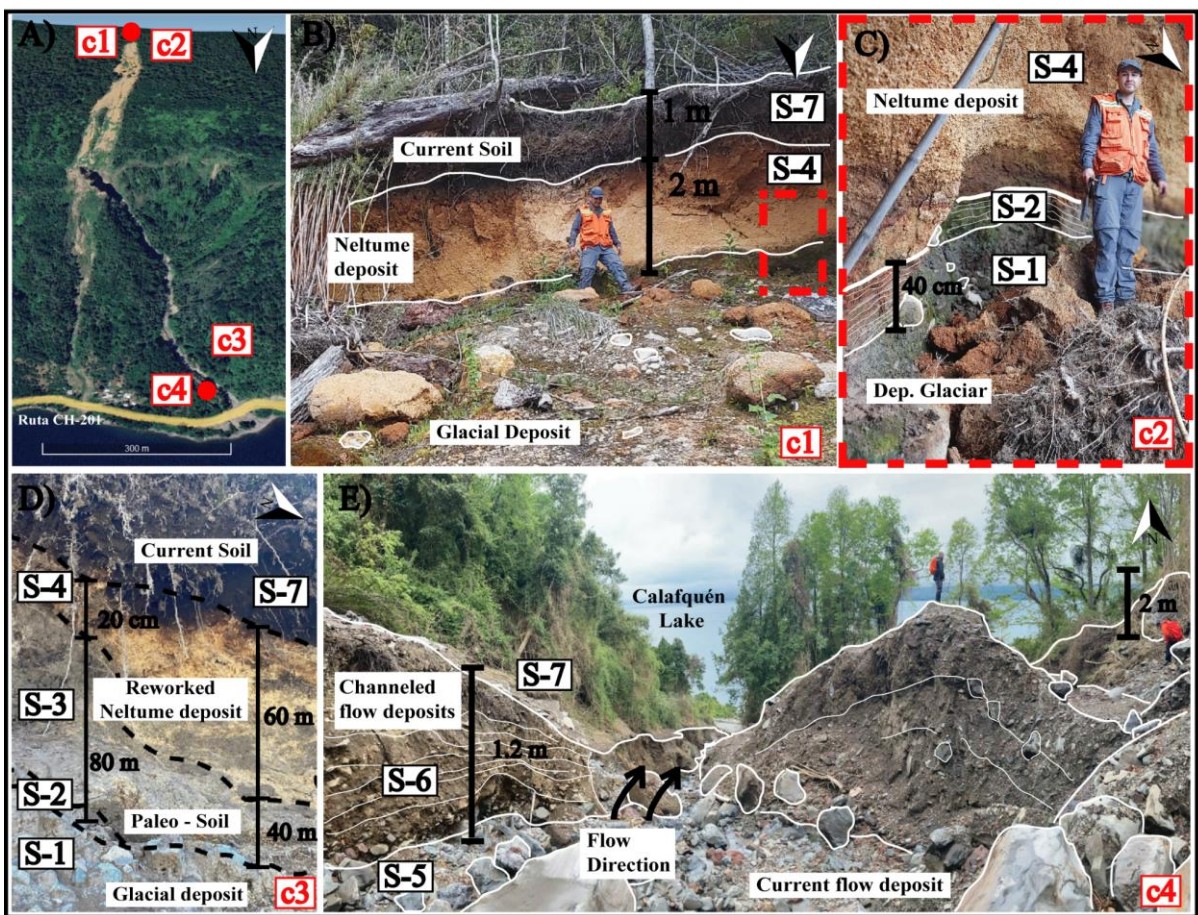

**Figure 4 Field photographs indicating soil deposits in scarp (B and C) and the deposits in the toe (inset D and E).**

Stratigraphic analysis along the slope (Figure 5B–D) reveals a consistent sequence overlying the CPfgr basement, characterized by a basal glacial deposit (S-1) with large angular clasts, overlain by silt–clay layers (S-2). These low-permeability units are covered by interbedded paleosols and volcanic ash-falls. In column C3, the paleosol S-3 was initially considered older than the Neltume ashfall, but the presence of reworked pyroclasts in S-4 (Figure 4D) suggests a younger relative age. The upper sequence concludes with active soil formation (S-7; Figure 5D).

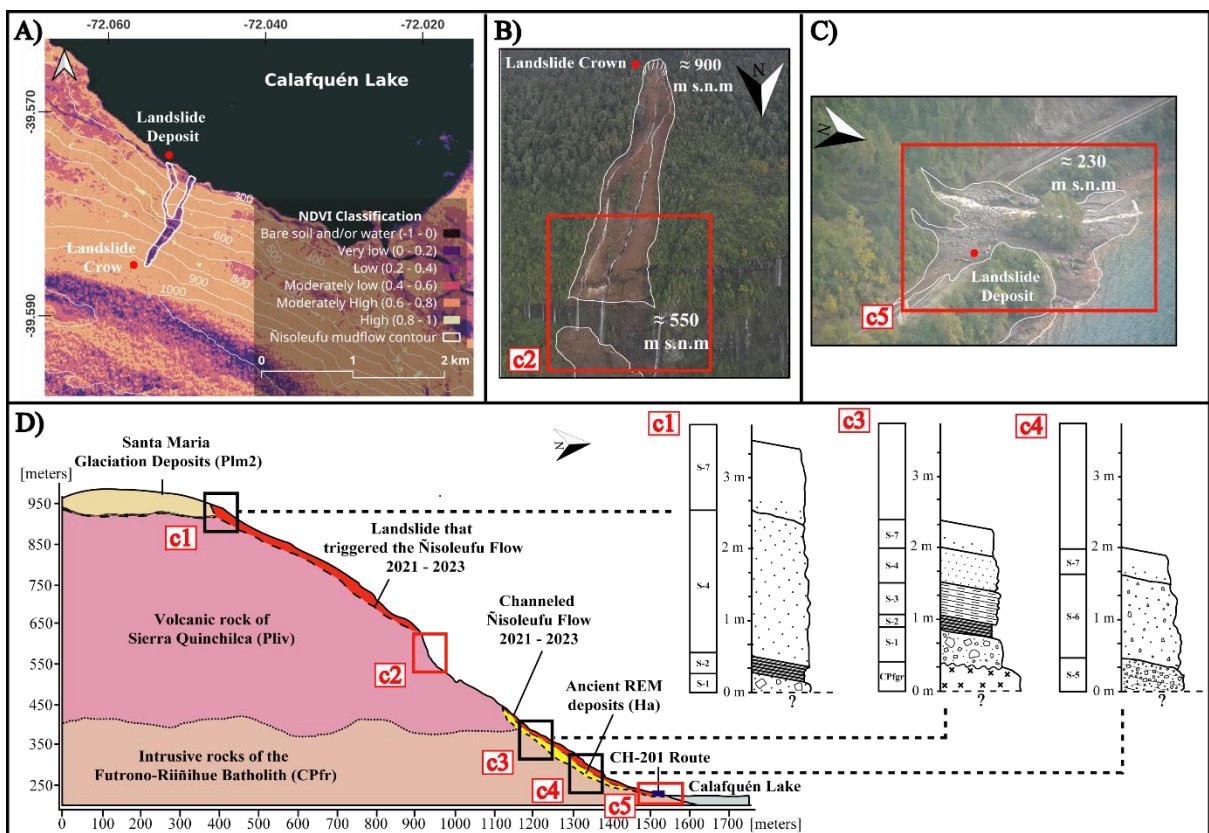

**Figure 5 A) Normalized Difference Vegetation Index (NDVI) highlighting the erosive zone generated by the Ñisoleufu debris flow. B) and C): Photos taken by a drone on 01 June 2021, highlighting the crown of the debris-flow (B), and the landslide deposit and the development of many waterfalls around the flow in red (C). D) Profile of the sequence where the flow occurred and stratigraphic columns. Photos: Carrasco and Ramirez (2021).**

Further downslope (250 m a.s.l.), column C4 exhibits a distinct stratigraphy dominated by mass-wasting deposits. The basal level (S-5) consists of a polymictic unit with sub-rounded clasts and pumice fragments from the Neltume event, embedded in a clay-rich matrix derived from upstream glacial units. A similar, though finer, deposit (S-6) overlies S-5, and is capped by the same modern soil unit (S-7). This stratigraphic framework—composed of alternating low-permeability substrates and reworked tephras—is representative of Andean terrains between 39° and 42°S and is critical to understanding hydrological storage and slope instability (Figure 1).

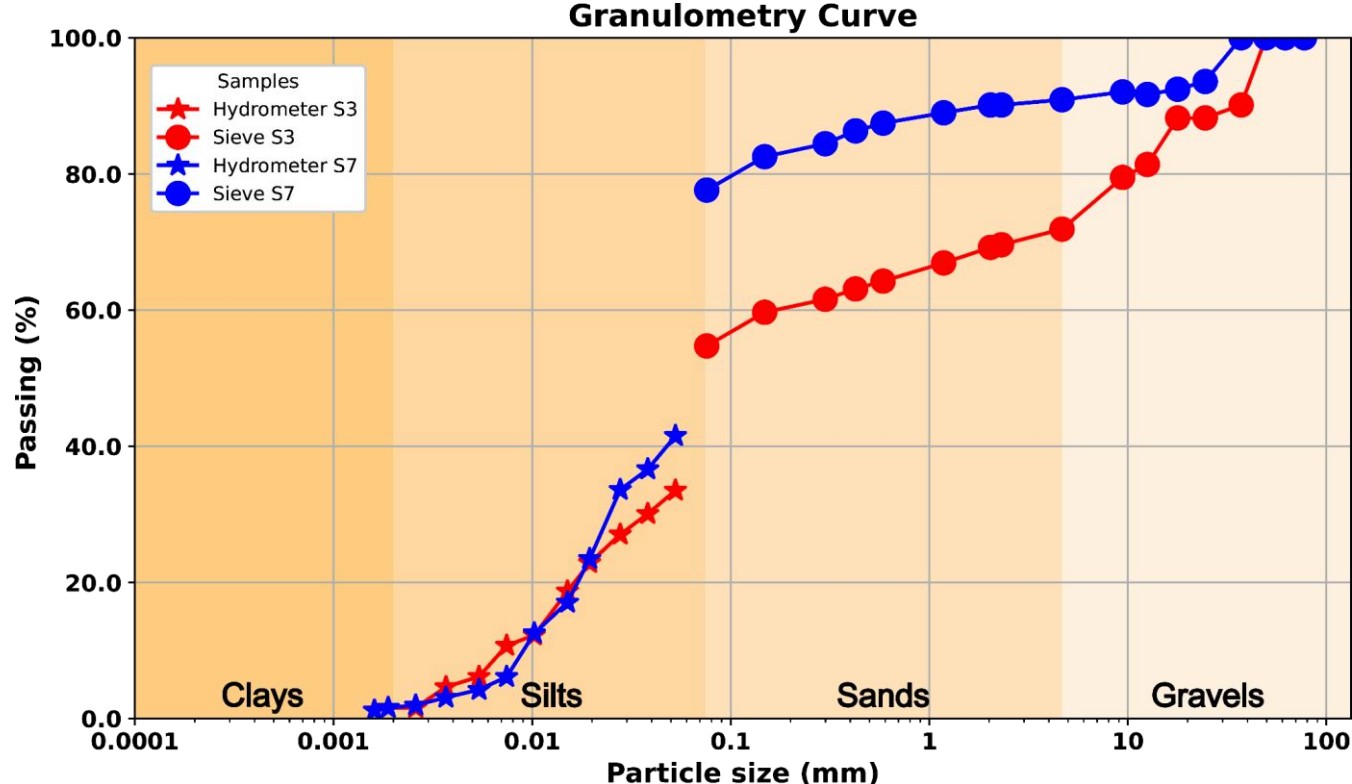

**Figure 6 Soil granulometric curve of S-3 and S-7.**

Table 2 Physical properties of soils related to the column C3.

| Soil type/Property | Normative | S-2 | S-3 | S-4 | S-7 |
|---|---|---|---|---|---|
| Moisture [w] (%) | NCh-1515 | 17.8 | 56.2 | 119.3 | 111.6 |
| Density [ρ] (g/cm³) | UNE-103-301-94 | 2.07 | 1.52 | <1 | 1.06 |
| Specific Gravity [$G_s$] | ASTM-D854-14 | 2.76 | 2.49 | 2.5 | 2.34 |
| Liquid Limit [$W_L$] (%) | AS 1289.3.9.1 | 27.48 | 123.93 | - | 149.83 |
| Plastic Limit [$W_P$] (%) | Nch 1517/2 | 16.07 | 91.3 | - | 114.13 |
| Plasticity Index [PL] | NCh1517/2 | 11 | 33 | - | 36 |
| Hydraulic Conductivity [$k_u$] (m/s) | Porchet and Laferrere (1935) | - | - | - | 3.13E-4 |

From a geotechnical perspective, the first soil (S-2) has a liquid limit of 27.48 and a plastic limit of 16.07, resulting in a

plasticity index of 11. This soil exhibits a low plasticity and is classified as a silt (CL) according to the Unified Soil

Classification System (USCS). S-3 showed a liquid limit of 123.93 and a plastic limit of 91.3, resulting in a plasticity index of

33. This soil exhibits a moderate plasticity and is classified as a plastic silt (CH), according to the USCS. Meanwhile, S-7 has

a liquid limit of 149.83 and a plastic limit of 114.13, resulting in a plasticity index of 36, showing a high plasticity and is

classified as a plastic organic soil (OH). The granulometric analysis (Figure 6; S-3 and S-7) confirms this classification, with
silt and sand content between 70% and 90%. Due to the soil properties, S-4 (Neltume ashfall pyroclastic) has not been
characterised in the laboratory due to the high fragility of the pyroclasts classified as lapilli (Ǿ>5 mm) being too coarse to
measure their limits.

## 4.2 Hydrometeorological conditions and precursory signals

The debris flow release zone (column c1 in Figure 4B **and** Figure 5B) provided accurate soil measurements using KSAT,
enabling us to understand the factors governing water transport on the slope. Field mapping revealed a layered model that
identified a highly stratified environment with distinct characteristics (Figure 4C; Figure 7). It was observed that the glacial
deposits underlying layers of volcanic origin regulate water movement in the soil due to their low hydraulic conductivity
(Figure 7). These glacial deposits, identified as moraines and varves, are associated with the Last Glacial Maximum (LGM),
correlated with nearby deposits and natural terrain conditions.

Notably, volcanic soil deposits (Figure 6; granular texture description) and Neltume ashfall demonstrating high saturated
hydraulic conductivity of 4.64E-5 to 3.31E-04 m/s (Figure 7). This characteristic is critical as it facilitates the movement of
infiltrated water from the organic surface to the slope's interior. In contrast, Varves-type glacial deposits show low hydraulic
conductivity, with values of 1.54E-05 m/s, while the till reached a Ks of 2.65E-05 m/s, both acting as partial barriers to water
movement. Nevertheless, their storage capacity is significant, particularly when situated on moraines that, due to their varied
granulometric distribution, retain water effectively. The hydraulic conductivity and the high infiltration rated derived from the
Porchet test showed a moderately high infiltrated rate (112.70 mm/hr), computing a k=3.31E-4 (m/s) in the superior organic
deposit layer (S-7; Table 2 **and** Figure 7).

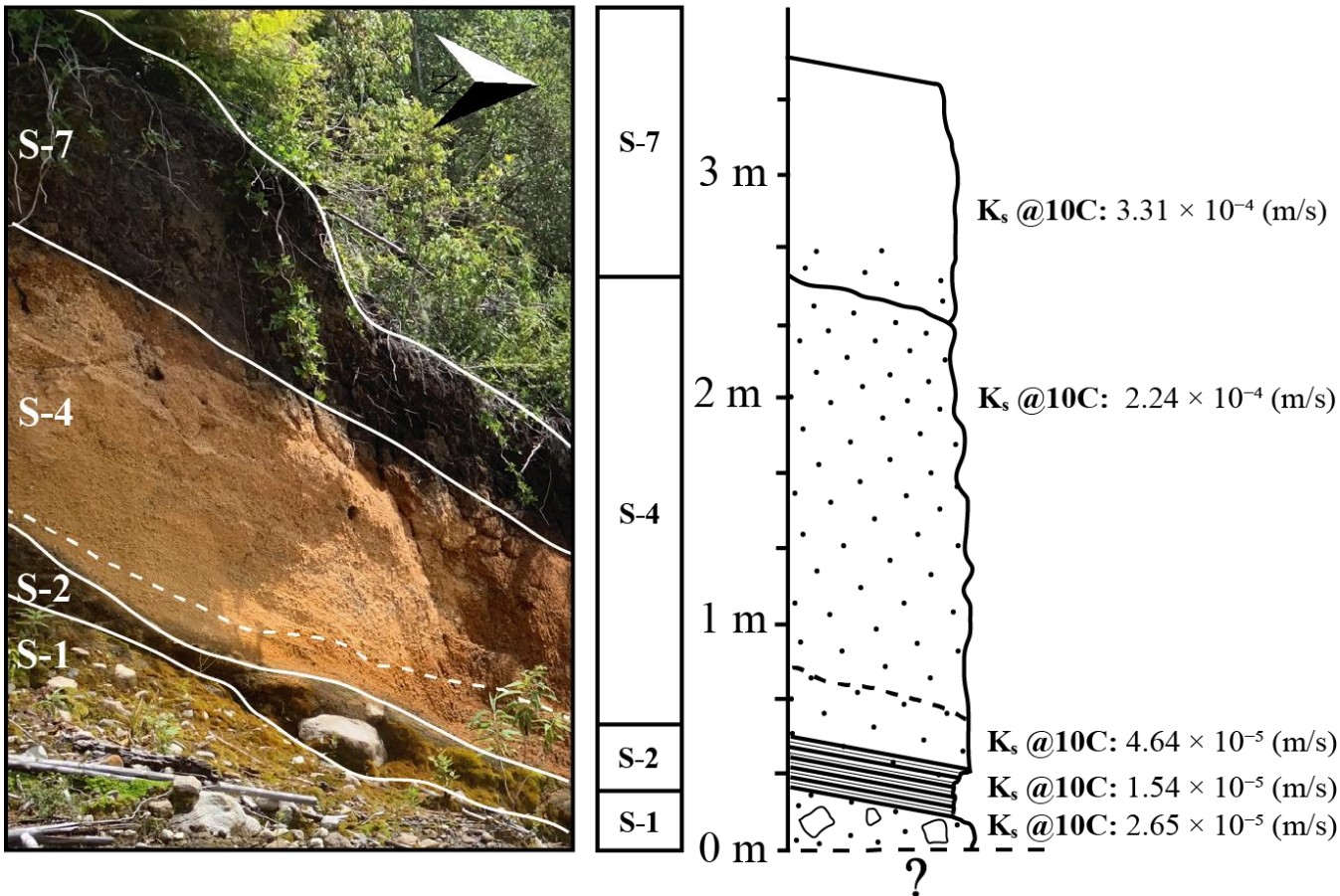

**Figure 7 Hydraulic properties of release zone in debris flow generation zone (column c1 in Figure 1).**

Analysis using the ERA5 model reveals a well-defined annual cycle in soil moisture (Figure 8A). While the overall evolution of moisture content does not indicate an anomalous trend when compared to previous years, significant short-term variations in soil moisture were observed (Figure 8B) in response to extreme precipitation events (Figure 8C). The preliminary results showed that antecedent rainfall led to surface saturation, with moisture levels reaching full saturation within the first metre of soil depth, primarily due to accumulated rainfall in the days prior (Figure 8B–C). Notably, a substantial and rapid increase in soil moisture was also detected at greater depths—up to 2.89 meters—where a 50% change occurred shortly before the onset of debris flow events (Figure 8B). This marked fluctuation was concentrated at the interface between tephra and till-varves, suggesting critical implications for slope instability. Our findings underscore that, even in the absence of a long-term anomalous trend, the combination of saturated soil conditions and intense rainfall plays a decisive role in triggering debris flows.

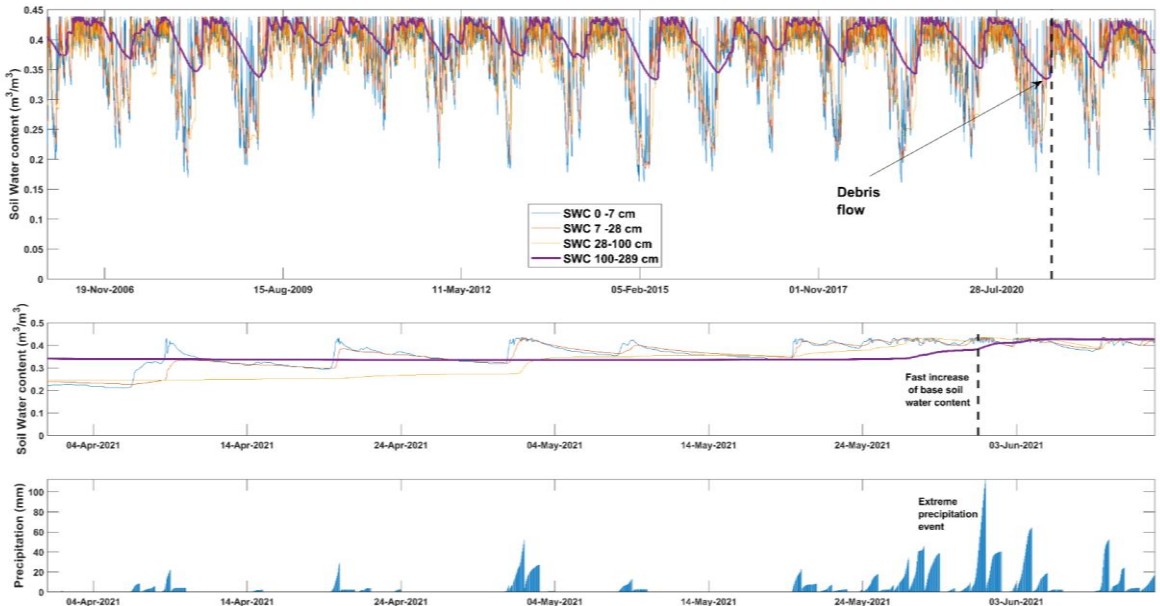

**Figure 8 Assessment of ERA5-land product over debris flow generation. A: Assessment of time series of soil moisture at different depths. B: Zoom to Soil moisture weeks previous to debris flow. C: Rainfall events accumulated at 24-hourly scale (ERA5-land).**

The availability of a sufficient quantity of SAR images and the revisit times of Sentinel-1 enabled a well-distributed temporal analysis of slope behaviour before the debris-flow event on May 31, 2021. We measured surface deformation between +9 and -32 mm/year. The results of PS estimation suggest the occurrence of surface subsidence consistent change in soil water content (Figure 8B). Our analysis reveals a precursory deformation signal that comprised two distinct phases (Figure 9). The first phase, beginning in late January 2021 and extending through April, exhibited a high-deformation pattern associated with consistent precipitation (Figure 8C), which infiltrated the soil and increased soil moisture levels (Figure 8B). This timing coincides with a significant precipitation event in late January, suggesting that summer rainfall may have acted as an initial trigger, initiating a surface deformation that evolved progressively over time. A second deformation phase following a high-intensity rainfall event in mid-to-late May 2021, characterized by pre-event deformation patterns. Despite that most evident precipitation event occurred in late May (15-day before), our surface deformation estimations could suggests that the triggering process likely began approximately in summer period. This temporal relationship between precipitation and deformation further underscores the need for incorporating hydrological factors into models of surface stability and landslide risk assessment. Finally, the deformation measurements are interrupted a few days before the debris flow occurs. This data interruption is attributed to the formation of significant extensional fractures (Figure 4D-E), causing a loss of coherence in the data (red box).

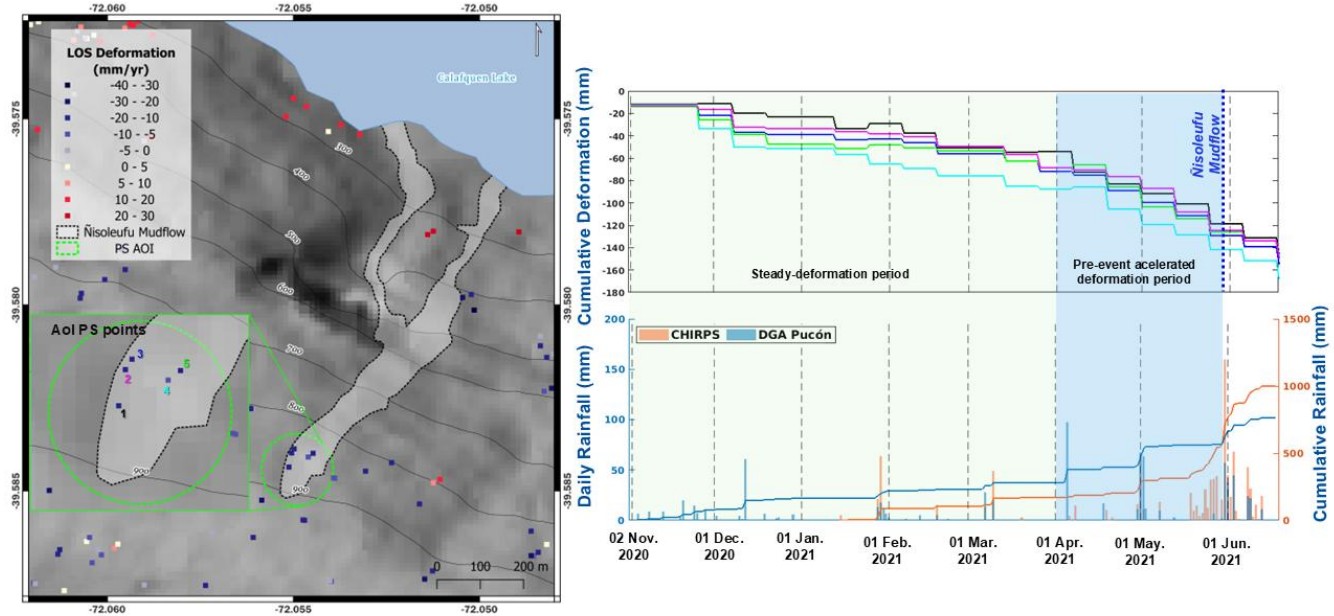

**Figure 9 PS points obtained with Sentinel-1 "ascending" orbit data before debris flow initiation. Precipitation time series using Pucón weather station. Deformation shows an subsidence in surface.**

## 5    Discussion

To improve debris-flow hazard assessment, it is crucial to understand how multiple controlling factors converge to trigger these events. In the Ñisoleufu case, we studied the triggering conditions of debris flow in an area to understand precursory signals of mass wasting initiation in post glacial and volcanic environments. We considered a geomorphological, geotechnical, hydrometeorological and surface deformation approach to constraint the variability of the mass wasting processes in one representative area of the Southern Andes.

### 5.1    Geomorphological and geotechnical implications

The occurrence of mass wasting events, such as landslides and debris flows, following periods of intense rainfall has been extensively studied in the Southern Andes using local cases, mainly based on the rainfall control over the mass wasting generation (Fustos et al., 2020; Maragaño et al., 2023) without consider the soil features as noted recently (Vasquez-Antipan et al., 2025). Our results showed that the geomorphology plays a crucial role in the generation of debris flows, particularly

through its influence on water accumulation in catchment areas, such as micro-basins (Fustos-Toribio et al., 2021). We considered two main geomorphological controls for debris flow initiation. First, stand out the steep slope (>45°), which was a significant contributing factor to the generation of the Ñisoleufu debris flow (Figure 2 **and** Figure 5B-D). Second, the northern orientation of the slope could also be a relevant factor, as in the central-southern Chile domain (36°-42°S), atmospheric moist flux from extreme rainfall tends to flow in a northwest (NW) direction, with orographic forcing triggering being enhanced by the Andean belt (Valenzuela and Garreaud, 2019; Vasquez-Antipan et al., 2025). The interplay between areas with high water accumulation capacity from local runoff and the slope's propensity to capture precipitation was instrumental in the generation of the debris flow. Stand out the slope that facilitated an efficient surface drainage of water from higher areas, leading to the formation of local accumulation and infiltration zones that promote the build-up of subsurface pore-water pressures (Figure 6B) leading to the debris flow event, facilitating a rapid downslope movement of soil once it became saturated.

The geotechnical stability of soils in the Southern Andes, could be controlled by the interaction between explosive volcanic. Mocho-Choshuenco volcanic complex (MCVC) and glacial processes and the resultant varved deposits (Figure 5) has been modulated the slope stability around the 39°S. The Neltume ashfall deposit related to the eruption of the MCVC (10,200+-500 BP; Rawson et al., 2015) played a significant role in the formation of the S-4 (Figure 4B-C). The thickness of all soils varies along the scar left by the debris flow, with the magnitude strongly influenced by the topographic gradient and/or erosion processes that might have occurred and following reworking (Figure 4D). Moreover, glacial deposit like as varves, formed from sedimentation during glacial periods, played a crucial role in the soil's physical and mechanical properties controlling the area with low hydraulic conductivity (Figure 7) introducing a water barrier in the area to regional scale.

Moreover, field evidence suggests that the Ñisoleufu event is not an isolated case, as indicated by other remobilised events in the area (Figure 2, geological map – alluvial deposit: Ha). The geotechnical properties of the remobilised materials are critical for defining slope stability conditions. Granulometric analyses indicate that the deposits are primarily granular soils, such as those associated with S-4, which are classified as frictional and are found overlying finer-grained cohesive soils, such as varves (S-2). Other soils found in the Southern Andes, including S-3 and S-7, originate from the decomposition of volcanic glass and glacial clays (Sanhueza et al., 2011; Vasquez et al., 2025), producing particles smaller than 0.1 mm (Figure 6).

Specifically, S-3 soils, derived from explosive eruptions of the Mocho-Choshuenco volcano, consist of non-cohesive volcanic ash mixed with fine-grained sediments, forming a matrix with elevated plasticity and a high liquid limit (Vasquez et al., 2025). These properties result from the introduction of fine material during the deposition. Moreover, S-7 soils, classified as organic soils derived from volcanic deposits, exhibit notably high liquid limits due to the accumulation of organic matter. The organic matter enhances the soil's water retention and promotes the formation of organic colloids, which may increase the liquid limit (Deng et al., 2017; Fiantis et al., 2019). Our results are consistent with independent laboratory testing in the zone (Vásquez et al., 2025), which shows that organic-rich paleosols were buried after the Last Glacial Maximum, approximately 5 km south of the study area, and exhibit similar liquid limit values to those observed in S-7.

The spatial distribution of soil layers varies abruptly along the slope, as observed in columns C1 and C3 for S-1, S-2, and S-4, indicating significant mass wasting and erosion processes near glacial lakes (Figure 5D). The frictional soils, such as those related to S-4, generally exhibit high shear strength (Chen et al., 2021), and when combined with steep topography, may contribute to the relative stability of post-glacial volcanic deposits (Walding et al., 2023; Ontiveros-Ortega et al., 2023). However, under extreme precipitation events—such as those recorded in recent years in the Southern Andes—soil saturation can substantially reduce the strength of even frictional soils, increasing the likelihood of failure (Fustos et al., 2017; Somos-Valenzuela et al., 2020; Fustos et al., 2021). This mechanism aligns with the observed extensional failures that preceded the initiation and reactivation of flows in June 2023 and 2024 (Figure 4C; Figure 10).

We propose that the event of Ñisoleufu, a classical case of the Southern Andes, was triggered by the soil saturation, reducing the effective stress within the soil matrix leading to extensional failure in the volcanic deposits (Figure 4B; Figure 10), highlighting the risks associated with this saturation, particularly in areas where explosive volcanic activity has previously occurred, leading to increased susceptibility to debris flows and other forms of mass wasting (Korup et al., 2019). The interaction between glacial deposits and volcanic materials creates a unique geotechnical environment where the stability of slopes is contingent upon both the physical properties of the soil and the hydrological conditions present.

## 5.2    Hydrometeorological conditions and precursory signals implications

The volcanic origin and composition of the soils evidenced a high soil moisture variability along the year. ERA5-land data reveals a sudden change in soil moisture content at various depths on the base of the debris flow initiation supporting the surface deformation days before the landslide (Figure 9). The water role in the event proposes that accumulation of subsurface water in granular soil could serve as a storage medium. The fine media of S-1 and S-2, with its low hydraulic conductivity, acts as barrier layer, enhancing the storage capacity of S-4 in the crown of the debris flow. This is supported by the hydraulic properties of S-3 (CH) and S-7 (OH), combined with Porchet's study on the current soil (S-7: 112.70 mm/hr), which indicates a moderately high infiltration rate and suggests high hydraulic conductivity for the Ñisolelfu soils (Figure 7). The combination of these factors contributes to the overall permeability of the volcanic soils, a main feature in the Southern Andes, potentially influencing water movement and retention in the area.

The water storage is consistent with surface deformation data, reaching -85 mm/yr in the previous 30 days of the event, offering insights of previous surface deformation to the occurrence of debris flow events in areas where glacial deposits and volcanic deposits coexist (Figure 9). The atypical nature of these deformations suggests that the surface shows a slow movement related to subsidence along LOS, followed by the debris-flow event on May 31, 2021 (159.8 mm/year). The constant slope deformation supports the hypothesis of the development of extensional failure, ultimately resulting in the observed a small landslide and subsequent release of water as debris flows (inset C in Figure 4; Figure 10).

The fast saturation of S-4 is exacerbated by increased water availability due to rainfall and by the presence of a bottom layer with low hydraulic conductivity associated with fine soils (S-2), being a common denominator in the Southern Andes. S-2 retains water and further reduces the shear strength of the underlying granular soil. Additionally, these soils could act as

lubricants, reducing the overall slope shear strength. The interaction between glacial deposits and explosive volcanic eruption

deposits in the cordilleran zone of the Southern Andes creates a scenario prone to slope deformation and mass removal,

especially debris flow. Our results suggest that the combination of geological and climatic factors in this region generates ideal

conditions for the occurrence of mass removal events under extreme precipitation events (Savi et al., 2016).

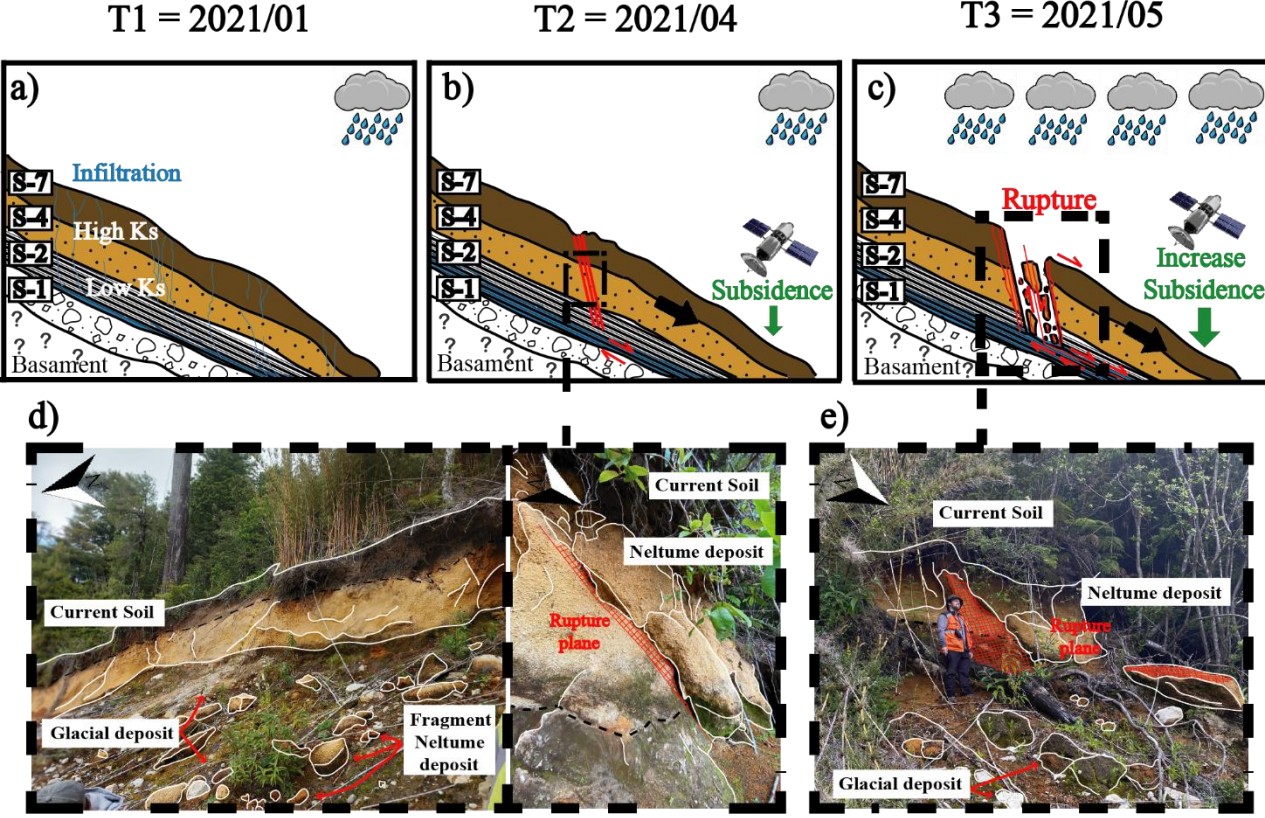

**Figure 10 Conceptual model of soil deformation and following failure under cryospheric constraint. A) First phase with low**
**precipitation. B) Start of the extensional failure and small-rate deformation measured by satellite. C) Failure and initiation of the**
**landslide and following debris flow. D) Scarp in the crown with extensional signatures (white lines) E) First plane of the current**
**rupture plane that could generate new debris mass wasting.**

Our results showed a limited amount of PS in the study area similar to previous studies in the area (Vasquez-Antipan et al.,

2025). The Southern Andes, and the Ñisoleufu area, is characterized by complex geomorphological features and varying

precipitation patterns that could introduce uncertainty to the remote sensing measurements. The application of limited

persistent scatterer data in assessing slope deformations offers a promising avenue for the development of a landslide early

warning system (LEWS) in the Southern Andes. However, the investigation into this method requires a thorough understanding

of potential limitations, particularly the extensive vegetation cover in the Southern Andes, which can obscure satellite signals

and affect data accuracy. Vegetation serves as a significant barrier to radar signals, leading to incomplete datasets that might

obscure important geological signals indicative of slope movements (Maragaño-Carmona et al., 2023). Therefore, additional efforts must be considered to move forward to an operational scale.

Climate change is increasingly debris flow generation by altering precipitation patterns and soil moisture dynamics (Talebi et al., 2007). In the Southern Andes, volcanic soils with variable textures play a critical role in this process. Enhanced seasonal moisture variability, exacerbated by extreme precipitation, leads to fast soil saturation, especially where fine-grained soils form low-permeability layers above coarser materials (Figure 10). These stratified soil conditions promote subsurface water storage, increasing the slope instability under saturated conditions (Talebi et al., 2007). Fine volcanic over glacial deposits can act as lubricants, further weakening slope cohesion and promoting failure (Espinosa et al., 2024) during intense rainfall, as happened during the Ñisoleufu event (Figure 7B). This event highlights how short-duration storms, increasingly associated with climate change, can overwhelm the buffering capacity of mountainous terrain. The soil media S-7 and S-4, both composed of organic-rich and granular volcanic materials, played a critical role in this response. During the 2023 event, infiltrating rainwater rapidly percolated through these coarse upper layers until reaching the underlying varved glacial sediments (S-2), which have significantly lower permeability. This layering caused a perched water table, increasing the pore pressure and reducing shear strength, ultimately contributing to slope failure. These effects were captured in our remote sensing observations, which showed expanded saturated zones and local instability near the contact between volcanic and glacial deposits.

Our results suggest strong stratigraphic controls and extreme precipitation events in the soils derived from volcanic materials overlying denser glacial layers, acting as failure planes under saturated conditions (Figure 7; Figure 8B). Our conceptual model promotes water retention and localized pressurization, especially during extreme rainfall events such as 2021 (not observed previously). The conditioning factors are further exacerbated by mid-term climate trends, including the ongoing megadrought in the Southern Andes (Garreaud et al., 2019), which increased desiccation cracking, and weakened root cohesion. Such drought-induced degradation lowers slope resistance, making even moderate precipitation more hazardous. Therefore, our observations suggest that the Ñisoleufu debris flow event, may exemplify the climate-induced changes in both hydrological extremes and landscape memory (drought legacy) act in concert to reduce slope stability. As extreme precipitation becomes more frequent under future climate scenarios, similar failures are expected to occur across a broader area than observed in past events (Figure 2A). Our findings stress the urgent need for debris flow forecasting models to incorporate stratified soil behavior, seasonal soil moisture dynamics, and drought-related weakening—factors essential to anticipating the growing hazard posed by climate change (Iverson et al., 2010; Gariano and Guzzetti, 2016).

## 5.3 Regional implications and Future Scope

The Southern Volcanic Zone (37.5° - 41.5°S; Stern 2004) has been shaped by significant volcanic Holocene eruptions (Fontaine et al., 2021; Moreno-Yaeger et al., 2024; Singer et al., 2024), leading to the formation of soils primarily composed of lapilli and/or ashes that settled following the contours of the topography, resulting in layers with varying thicknesses (Stern,

2007). Old deposits in the zone show similarities to the current debris flow deposits (Figure 5B; Figure 10A), suggesting that the event triggered in 2021 is not an isolated occurrence in the area. This evidence points to a history of recurring debris flow events in the region. This condition is dominant in all the surroundings of the Mocho-Choshuenco volcanic complex (Rawson et al., 2015; Moreno-Yaeger et al., 2024).

The Ñisoleufu debris flow showed a characteristic pattern of mass wasting processes in the Southern Andes, becoming analogues to Petrohue event (Fustos-et al., 2021) in Osorno Volcano (Figure 1D). The occurrence of debris flows in the Southern Andes, particularly in the border of the maximum extension of the Patagonian ice sheet suggest that debris flows are typically related to saturated soils that can transform into more fluidic mixtures. Our observations and previous studies (Davies et al., 2020; Fustos et al., 2021) proposes a strong correlation between debris flows and glacial moraines, particularly where the last glacial maximum shaped the relief (Figure 1B and Figure 1C). The interaction of past glacial dynamics with contemporary environmental processes related to new precipitation patterns provides a backdrop for increased debris flow activities in Southern Andes based on rainfall-induced mass wasting data from Fustos et al. (2022). Stand out regions at the borders of these past glaciers tend to exhibit increased susceptibility to debris flow generation due to the combination of steep slopes and the prevalence of loose moraine deposits (Figure 8B and C), which can be mobilized during significant precipitation events (Sepúlveda et al., 2014).

Our results show that precursory signals such as small progressive deformation of the surface (Figure 10) could suggest the indication of possible soil slides previous to initiation of the debris flow (Figure 10). This phenomenon is influenced by fine soil interspersed with glacial deposits, superimposed on moraines and volcanic deposits, common throughout the southern Andean region (Moreno-Yaeger et al., 2024; Singer et al., 2024). A simple geomorphological generalization highlights the necessity of investigating the interaction between glacial and volcanic deposits to understand better the interplay between media with constraints on soil hydraulic conductivity. Future studies should prioritise examining these dynamics to prevent and mitigate potential risks associated with similar landslide events across the broad area of the Southern Andes.

Debris flow generation in volcanic soils and their correlation with post-glacial eruptions, assumes a critical importance in comprehending the mass wasting hazards under the current climate crisis. Holocene eruptions, known for their Plinian events such as Mocho-Choshuenco, Carran-Los Venados, Calbuco, Chaiten in the Southern Andes (Singer et al., 2024), have yielded abundant tephra and volcanic soils possessing high hydraulic conductivity. These eruptions formed volcanic deposits over moraine and varve deposits (Figure 4). The volcanic soil acts as a high-rate infiltration layer meanwhile, the moraine crowned by impermeable varve layers in the base of the sequence plays a reservoir on (Figure 10). Therefore, the amalgamation of these deposits with intense precipitation events may expedite the loading of these reservoirs, heightening the probability of slope instability due to increased mechanical load and pore-stress changes (Bogaard & Greco, 2015). Early indications of this phenomenon are evident in occurrences such as Chocol 2023, and Volcán Osorno (every year), suggesting a potentially heightened recurrence in the future. To mitigate the associated risks effectively, it is imperative to enhance the zoning of areas

prone to debris flows in the Southern Andes, thus minimizing the impact on population and infrastructure within these vulnerable zones.

On a regional scale, climate change is intensifying debris flow hazards worldwide (Gariano and Guzzetti, 2016). In the Southern Andes, current changes in precipitation patterns will affect the stability of volcanic and glacial deposits through alterations in water storage, as noted in this study (Figure 8B). The region's unique stratigraphy in South America, where volcanic soils overlay glacial sediments, may become unstable during extreme rainfall events. Significant shifts in precipitation patterns, as predicted by CMIP6 models, alter the spatial distribution of precipitation and their impact on soil moisture storage, with limited accurate estimations (Salazar et al., 2023). We propose that future developments should carefully constrain areas with high susceptibility to debris flow. Therefore, improved hazard debris flow delimitation and instrumental monitoring become critical for reducing the impact of these hazards in the Southern Andes.

Our study contributes to understanding the relationship between volcanic and glacial deposits under extreme rainfall events forcing in the Southern Andes. However, further research is necessary to improve the RIL susceptibility models due to an incomplete integration of critical soil properties. Several studies focus on rainfall thresholds as primary trigger, oversimplifying the failure mechanisms. Our study proposes that additional assessment of the hydraulic and mechanical influence of specific soil layers must be considered. Additionally, the pronounced spatial heterogeneity in soil layer composition in Southern Andes, ranging from S-1 to S-7, and variability in thickness and permeability, further complicates predictive accuracy to regional scale. Future developments must consider high-resolution subsurface mapping introducing national scale models (Dinamarca et al., 2024), allowing better RIL risk models overlooking zones where possible saturated soil could appear, leading to sudden failure. Moreover, surface deformation frequently observed as a slow extensional signal prior to collapse introduces a limited amount of PS. Moreover, the hydrometeorological variability indicates that better soil moisture models are necessary in the zone to improve the slope stability analysis. Together, these limitations highlight the urgent need for multidisciplinary approaches that integrate geotechnical, geomorphological, and hydrometeorological data into landslide hazard assessments.

Our study proposes considerations that must be assessed in landslide research in the Southern Andes, allowing a deep understanding of the interplay between geomorphological, geotechnics, hydrometeorological and remote sensing analysis, allowing a better landslide risk assessment and mitigation efforts in the region. We propose that the Ñisoleufu event could as study case for researchers and authorities can better comprehend the specific conditions that lead to debris-flow occurrences and implement appropriate measures to minimize the impact of such events in the future. Finally, these results suggest that hazard assessment protocols should include hydrogeotechnical mapping of tephra-over-moraine systems, combined with seasonal soil moisture dynamics and weather forecasts of extreme rainfall as early indicators of instability in these settings.

# 6    Conclusions

We studied the conditions that evolved in the generation of debris flows in the Southern Andes, an area modulated by the glacial and volcanic processes. The mass wasting could be influenced by complex interactions among geomorphological,

geotechnical, and hydrometeorological factors (Figure 10). The geological environment of the Southern Andes, characterized by a mix of volcanic and glacial deposits, showcases unique soil properties affecting overall slope stability. The mechanisms leading to mass wasting, particularly the Ñisoleufu event, underscore the critical roles of soil saturation and effective stress reduction in triggering slope failures (Figure 8). The analysis of geomorphological factors revealed that slopes greater than 30 degrees significantly contribute to debris flow triggers, supporting the emphasis of rainfall's role in mass wasting (Figure 5). The orientation of these slopes, particularly aspects with more rainfall exposure, promotes precipitation accumulation from extreme weather patterns. Our findings corroborate the influence of gradual accumulation of subsurface water, aided by low hydraulic conductivity from underlying soil layers (Figure 7), can create critical saturation levels that reduce shear strength and lead to flow initiation. These processes, not considered in detail must be integrated in future assessment in detail into the future.

We conclude that recurrence of mass wasting is influenced by past volcanic eruptions and post-glacial conditions. A comprehensive understanding of the interplay between geological and hydrometeorological conditions is crucial for forecasting debris flow risks in the Southern Andes, particularly considering climate change, which may exacerbate extreme weather events. We highlight that while frictional soils can provide stability under dry conditions, they become increasingly vulnerable to failure under saturated conditions as was evidenced in the Ñisoleufu event. The data indicating surface deformation prior to debris flow events (Figure 9), demonstrating that precursory signals that can be leveraged for early warning systems. The velocity of surface movements preceding the Ñisoleufu event are correlated with increased soil moisture levels, emphasizing the utility of integrating remote sensing technologies to monitor these changes as proxy. Our findings could be extended to regional scale as a conceptual model of the landslides and debris flows in the Southern Andes, suggesting areas prone to such occurrences and should be monitored closely to develop effective mitigation strategies. Finally, our results demonstrate the value of integrating geomorphological, hydrometeorological, and hydrogeotechnical data to support debris-flow hazard assessments. In particular, the combination of tephra layers overlying low-permeability glacial deposits, together with rapid water infiltration on steep slopes during extreme rainfall events, defines a critical configuration that enhances susceptibility to failure. This integrated framework provides a robust basis for identifying high-risk areas and strengthening early warning strategies in the Southern Andes and comparable volcanic-glacial settings worldwide.

## 7      Code availability

All the codes used in this manuscript are reproducible from the main text. We can deliver the main script under any request.

## 8 Data availability

The datasets used in this study are available in the paper. The ALOS-PALSAR DEM is publicly accessible at https://search.asf.alaska.edu/#/?dataset=ALOS (last access: 30 September 2024). Additional information about the information or datasets can be obtained from Ivo Janos Fustos-Toribio, ivo.fustos@ufrontera.cl.

## 9 Author contributions

IF, DB, AB, GF and AG contributed to the conceptualisation and methodology of the research and performed the formal analysis, visualisation and validation. IF and DB were involved in the funding and supervision of the paper. IF and AB contributed with the supervision, review and editing of the paper. SS and JLP contributed to the discussion of the scientific results. All the authors provided input in terms of methodology and the review and editing of the paper.

## 10 Competing interests

The authors declare that they have no conflict of interest.

## 11 Disclaimer

## 12 Acknowledgements

This work was made possible thanks to the "Agencia Nacional de Investigación y Desarrollo (ANID)" of the Chilean Government; "Fondecyt Regular" grant 1230792); "Fondecyt post-doctoral" grant 3200387) and CIVUR-39° "Centro Interactivo Vulcanológico de La Araucanía" Project UFRO2193 of the Desarrollo de Actividades de Interés Nacional (ADAIN), Ministry of Education, Chilean Government. We appreciate the support of Mauricio Hermosilla for their support in geotechnical analysis.

## 13 Financial support

This research has been supported by ANID (grant nos. Fondecyt 3200387 and Fondecyt 1230792). Laboratory instruments were supported by FONDEF ID23i10118.

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
