# Peer review of "Controls over debris flow initiation in glacio-volcanic environments in the Southern Andes."

_EGUsphere, 2025_

## Author Comment (AC1)

**Controls over debris flow initiation in glacio-volcanic environments in the Southern Andes**

Response of authors

**Reviewer #1:**

**General comments:**

This manuscript presents a comprehensive investigation into the initiation mechanisms of debris flows in the Southern Andes, with a focus on the Ñisoleufu event. The authors attribute this event to increasing soil saturation during an extreme rainfall episode under typical glacio-volcanic geological conditions, where a low-permeability basal layer promotes water retention in the overlying soil.

RC1_1: However, the extent to which this case represents common conditions across the Southern Andes, as implied by the title, has not been adequately discussed.

> **A:** We agree. A new Figure 1 showing the emplacement of rainfall-induced landslides and debris flow was introduced. Also, new text was introduced in the Discussion section 5.3, "Regional implications and future scope".

> **Figure 1** Rainfall-Induced mass wasting in Southern Andes. A) Regional map of Ice-sheet extension during 35 ka and 20 ka as example and volcanoes emplaced in the area. B) Zoom to study area with Ñisoleufu in Northen Ice sheet sector showing the weather stations. C) Zoom to Northern Patagonian area showing correlation between mass wasting events and moraine lines (blue line). D) Zoom to Osorno volcano area showing high debris flow generation area discussed in Fustos et al., 2022.

[Figure]

These figure was discussed in detail in the subsection 5.3

Additional paragraph:

The Ñisoleufu debris flow showed a characteristic pattern of mass wasting processes in the Southern Andes, becoming analogues to Petrohue event (Fustos-et al., 2021) in Osorno Volcano (Figure 1D). The occurrence of debris flows in the Southern Andes, particularly in the border of the maximum extension of the Patagonian ice sheet suggest that debris flows are typically related to saturated soils that can transform into more fluidic mixtures. Our observations and previous studies (Davies et al., 2020; Fustos et al., 2021) propose a strong correlation between debris flows and glacial landforms, particularly where the last glacial maximum shaped the relief (examples in Figure 1B and Figure 1C). The interaction of past glacial dynamics with contemporary environmental processes related to new precipitation patterns provides a backdrop for increased debris flow activities in Southern Andes based on rainfall-induced mass wasting data from Fustos et al. (2022). Stand out regions at the borders of these past glaciers tend to exhibit increased susceptibility to debris flow generation due to the combination of steep slopes and the prevalence of loose moraine deposits (Figure 1B and C), which can be mobilized during significant precipitation events (Sepúlveda et al., 2014).

RC1_2 Additionally, the overall structure and clarity of the manuscript require improvement, which currently undermines the value of the research. Further comments are detailed below.

**A:** We appreciate and agree with the reviewer's comments, focusing on his specific comments and making the suggested changes with the goal of improving the manuscript.

**Specific comments:**

1. **SC1_1 L32:** The opening sentence suggests that debris flows may result in extreme rainfall, which is likely a typographical error. Please carefully review the manuscript for similar errors.
   - **A:** We agree with R1 about the inconsistency of this idea. It was changed to: "*Episodes of extreme rainfall have increased due to climate change, resulting in a greater frequency of debris flows...*"

2. **SC1_2 Introduction:** The introduction requires restructuring. A clearer overview of the Southern Andes in terms of debris flow features and glacio-volcanic environments is needed. The current structure lacks coherence; for instance, the paragraph starting at Line 41 introduces the importance of understanding debris flows in inhabited regions but abruptly shifts to discussing the impact of climate change. At the same time, please reduce repetitive content about the research's significance.
   - **A:** We agree with reviewer 1 and 2. We have addressed all the requested modifications in our response to comment RC2_2. Now, we rewrite the introduction, introducing a paragraph 3 where we discuss the state of art in Southern Andes under glacio-volcanic point of view.

     **New paragraph:**

     Nowadays, understanding the impact of debris flows in glacial environments become critical in the Chilean southern Andes, particularly due to the most part of the inhabitants lives there. Changes of precipitation patterns related to climate change, particularly fast and intense rainfall events, could amplify the frequency and magnitude of debris flows (Fustos et al., 2022). An increase of extreme hydrometeorological events affecting slopes in glacial settings is observed, whose mechanical properties and geomorphology have evolved since the Last Glacial Maximum (Fustos-Toribio et al., 2021b; Somos-Valenzuela et al., 2020; Ochoa-Cornejo et al. 2025). Considerable uncertainty remains about how the interaction between volcanic-derived soils over glaciar landforms will respond to extreme hydrometeorological events. One of the current model initiation of debris flow are related to slow deforming surface in hillslopes stand out, mainly due to gravity and surface erosion during high precipitation events (Xie et al, 2020; Yi et al, 2021). Slow surface deformation could lead to extensional failures that could expand and deepen, generating landslides and evolving into debris flows, especially under water-saturated conditions or heavy rainfall (Gregoretti, 2000; Fustos et al., 2017; Wang et al., 2024). The capacity to oversee these extensional failures in remote areas close to roads is an open question yet, mainly in Southern Andes.

3. **SC1_3 L104:** The authors mention that three complementary methodologies were used, yet only two are listed in the introductory paragraph of the methodology section.

   ○ **A:** Our apologies, is an error. Now we improved the redaction.

   **Original text:**

   To achieve this, we employed three complementary methodologies.

   **Modified text:**

   To achieve this, we employed two complementary methodologies.

4. **SC1_4 Figure 4D:** It is recommended to display the position of the profile line in a plan view (e.g., in Figure 4A). Additionally, please consider replacing the north arrow with one indicating the slope direction angle, which would aid reader comprehension.

   **A:** We agree with the reviewer's comment. To improve the presentation of Figure 5A (previously Figure 4A), we have modified it to include the position of the profile line, labelled as AA′. This identifier also appears in Figures 2D and 2A (previous Figures 1D and 1A) to ensure consistency and facilitate interpretation. Additionally, we have removed the north arrow from Figure 5D and replaced it with an indication of the slope direction, aligned with the AA′ profile. This change is intended to enhance the reader's understanding of the topographic context.

   Original Figure 5

[Figure]

Modified Figure 5

[Figure]

Original Figure 2:

[Figure]

Modified Figure 2:

[Figure]

5. **SC1_5** L146: Data from four weather stations were used, but only the Pucon station is plotted in the figure. Where is this station located and why do you select it as representative?
   - **A:** For the analysis of the event that occurred in the Ñisoleufu area, we exhaustively reviewed the meteorological stations located nearby. A total of four stations were identified within a reasonable radius for potential use as sources of meteorological data. However, upon examining the temporal

coverage and continuity of their records, it was found that only the Pucón station had complete and operational data for the study period. Panguipulli, Conaripe and Puesco had records ending in 2019, while others exhibited data gaps during the Nisoleufu months, probably due to telemetry failures. These interruptions compromised the use of the data required for the analysis, and thus, these stations were excluded. Therefore, the use of the Pucón station is fully justified as the most reliable and representative source of meteorological data for this study. Now, we included in a new figure the location of the weather stations as Figure 1.

New Figure:

**Figure 1** Rainfall-Induced mass wasting in Southern Andes. A) Regional map of Ice-sheet extension during 35 ka and 20 ka as example and volcanoes emplaced in the area. B) Zoom to study area with Ñisoleufu in Northen Ice sheet sector showing the weather stations. C) Zoom to Northern Patagonian area showing correlation between mass wasting events and moraine lines (blue line). D) Zoom to Osorno volcano area showing high debris flow generation area discussed in Fustos et al., 2022.

[Figure]

The same information was introduced in the text:

**Original text in section 3.3:**

We specifically selected data from the Pucón station and CHIRPS dataset for comparison with the time series of deformation to examine the relationship between precipitation and deformation.

**Modified text in section 3.3:**

The four weather stations were identified within a reasonable radius for potential use as sources of meteorological data. However, upon examining the temporal coverage and continuity of their records, it was found that only the Pucón station had complete and operational data for the study period.

Moreover, we merged this data with CHIRPS dataset for comparison with the time series of deformation to examine the relationship between precipitation and deformation.

6. **SC1_6** Furthermore, based on this data, the debris flow appears linked to intense rainfall approximately 15 days prior to the event. Why is the pre-event described as occurring two months earlier?

> **A:** We appreciate the reviewer's insightful comment. We agree that the final triggering phase of the event appears to have occurred approximately 15 days prior to the main manifestation. However, based on our analysis of InSAR time series (Figure below), we identified changes in surface deformation beginning in late January. This timing coincides with a significant precipitation event during the same period.

[Figure]

> We therefore propose that the summer rainfall may have acted as an initial trigger, initiating a lateral spreading process that evolved progressively over time. In this context, we consider the event to be the outcome of a process that began during the austral summer precipitation, even though its most evident expression occurred later.

> **Original text:**

>> Our results suggest a precursory deformation signal based on two distinct periods (Figure 7). The first period displayed a high-deformation pattern starting from January 2021 until April 2021 marked by consistent precipitation (Figure 6C) that infiltrated into the soil increasing the soil moisture (Figure 6B). A second period characterized by pre-event deformation patterns following a high precipitation event in May 2021. Our results suggest that the variation in surface deformation rates could be a response to a significant precipitation event that occurred at the end of January, being related to the high-intensity rainfall throughout the second half of May (Figure 7).

**Modified text:**

Our analysis reveals a precursory deformation signal that comprised two distinct phases (Figure 9). The first phase, beginning in late January 2021 and extending through April, exhibited a high-deformation pattern associated with consistent precipitation (Figure 8C), which infiltrated the soil and increased soil moisture levels (Figure 8B). This timing coincides with a significant precipitation event in late January, suggesting that summer rainfall may have acted as an initial trigger, initiating a surface deformation that evolved progressively over time. A second deformation phase following a high-intensity rainfall event in mid-to-late May 2021, characterized by pre-event deformation patterns. Despite that most evident precipitation event occurred in late May (15-day before), our surface deformation estimations could suggest that the triggering process likely began approximately in summer period.

7. **SC1_7** Table 2: Both plastic and liquid limits of samples S-3 and S-7 are exceptionally high, approaching or exceeding 100%, which is unusual in geotechnical testing. Have these results been validated? If so, the unique nature of these soil layers should be highlighted and discussed in detail.

**A:** We agree with the reviewer's observations and have thoroughly reviewed both the methodologies employed and the results obtained, reaffirming the high Atterberg limits presented in Table 2. These values were carefully measured using standard procedures (AS 1289.3.9.1 and NCh 1517/2), validating the results as being consistent with previous results obtained from the same soils (Vasquez et al., 2025). Following the reviewer's suggestion, we have now included a more detailed discussion in the revised manuscript (Section 5.2), highlighting the distinctive geotechnical behaviour of these soils. The exceptionally high plastic and liquid limits observed in samples S-3 and S-7 likely reflect the volcanic soil genesis, weathering, and extreme climate processes around the Mocho-Choshuenco volcanic complex. This singular pedogenetic pathway—characteristic of glacio-volcanic terrains in the Southern Andes—produces fine-grained, organic-rich soils with high water retention and deformability, which in turn play a critical role in slope stability and debris flow initiation in the region.

**Original text in section 5.2:**

Moreover, field evidence suggests that the Ñisoleufu event is not an isolated case as seen in the remobilised events (Figure 1, geological map - alluvial deposit: Ha). The geotechnical properties of the material to be remobilised are crucial for establishing stability conditions. The granulometric characteristics of the deposits, primarily granular types associated with S-4, are identified as frictional soils overlying fine-grained, cohesive soils like varves (S-2).

Other soils in Southern Andes, such as S-3 and S-7, could be originated from the decomposition of volcanic glass from ashes and glacial clays (Sanhueza et al., 2011), resulting in particles smaller than 0.1 mm (Figure 5). The distribution of the soil layers varies abruptly downslope, as observed in columns c1 and c3 for S-1, S-2, and S-4, indicating intense mass wasting and erosion productivity in areas close to glacial lakes (Figure 4D). The frictional soils, related to S-4, exhibit high shear resistance (Chen et al, 2021), combined with steep slopes, can contribute to stability control of post-glacial volcanic deposits (Walding et al, 2023; Ontiveros-Ortega et al, 2023). However, while frictional soils are generally more resistant to sliding (Chen et al, 2021), soil saturation can significantly decrease their strength, thus increasing the risk of failure under extreme precipitation events detected in recent years in the Southern Andes (Fustos et al., 2017; Somos-Valenzuela et al., 2020; Fustos et al., 2021). This is consistent with the presence of extensional failure observed before flow initiation and subsequent reactivations in June 2023 and 2024 (Figure 3C; Figure 8).

**Modified text in section 5.2:**

Moreover, field evidence suggests that the Ñisoleufu event is not an isolated case, as indicated by other remobilised events in the area (Figure 2, geological map – alluvial deposit: Ha). The geotechnical properties of the remobilised materials are critical for defining slope stability conditions. Granulometric analyses indicate that the deposits are primarily granular soils, such as those associated with S-4, which are classified as frictional and are found overlying finer-grained cohesive soils, such as varves (S-2). Other soils found in the Southern Andes, including S-3 and S-7, originate from the decomposition of volcanic glass and glacial clays (Sanhueza et al., 2011; Vasquez et al., 2025), producing particles smaller than 0.1 mm (Figure 6).

Specifically, S-3 soils, derived from explosive eruptions of the Mocho-Choshuenco volcano, consist of non-cohesive volcanic ash mixed with fine-grained sediments, forming a matrix with elevated plasticity and a high liquid limit (Vasquez et al., 2025). These properties result from the introduction of fine material during the deposition. Moreover, S-7 soils, classified as organic soils derived from volcanic deposits, exhibit notably high liquid limits due to the accumulation of organic matter. The organic matter enhances the soil's water retention and promotes the formation of organic colloids, which may increase the liquid limit (Deng et al., 2017; Fiantis et al., 2019). Our results are consistent with independent laboratory testing in the zone (Vásquez et al., 2025), which shows that organic-rich paleosols were buried after the Last Glacial Maximum, approximately

5 km south of the study area, and exhibit similar liquid limit values to those observed in S-7.

The spatial distribution of soil layers varies abruptly along the slope, as observed in columns C1 and C3 for S-1, S-2, and S-4, indicating significant mass wasting and erosion processes near glacial lakes (Figure 5D). The frictional soils, such as those related to S-4, generally exhibit high shear strength (Chen et al., 2021), and when combined with steep topography, may contribute to the relative stability of post-glacial volcanic deposits (Walding et al., 2023; Ontiveros-Ortega et al., 2023). However, under extreme precipitation events—such as those recorded in recent years in the Southern Andes—soil saturation can substantially reduce the strength of even frictional soils, increasing the likelihood of failure (Fustos et al., 2017; Somos-Valenzuela et al., 2020; Fustos et al., 2021). This mechanism aligns with the observed extensional failures that preceded the initiation and reactivation of flows in June 2023 and 2024 (Figure 4C; Figure 10).

○ **References:**

Deng, Y., Cai, C., Xia, D., Shuwen, D., Chen, J., & Wang, T. (2017). Soil atterberg limits of different weathering profiles of the collapsing gullies in the hilly granitic region of southern china. Solid Earth, 8(2), 499-513. https://doi.org/10.5194/se-8-499-2017

Fiantis, D., Ginting, F., Gusnidar, G., Nelson, M., & Minasny, B. (2019). Volcanic ash, insecurity for the people but securing fertile soil for the future. Sustainability, 11(11), 3072. https://doi.org/10.3390/su11113072

Ustiatik, R., Ariska, A., Hakim, Q., Wicaksono, K., & Utami, S. (2023). Volcanic deposits thickness and distance from mt semeru crater strongly affected phosphate solubilizing bacteria population and soil organic carbon. Journal of Ecological Engineering, 24(10), 360-368. https://doi.org/10.12911/22998993/170860

8. **SC1_8** Figure 6: The ERA5-based analysis reveals a clear annual cycle in soil moisture content. However, similar levels of high soil moisture appear to have occurred in previous years, such as around 5 February 2015. What additional factors, beyond soil moisture, may explain the occurrence of the 2023 event compared to those previous intervals?

A: We appreciate the reviewer's comment regarding the need to explore additional factors beyond soil moisture that may explain the occurrence of the 2023 debris flow event, particularly when similar soil moisture levels were observed in previous years (e.g., February 2015) without resulting in such events. We agree with this observation and now we have elaborated a

more comprehensive explanation in the revised manuscript. Now, we added in 5.2 two new paragraphs:

Climate change is intensifying debris flow hazards by increasing the frequency and severity of extreme precipitation events and disrupting soil moisture regimes (Talebi et al., 2007). In the Southern Andes, these changes interact with stratified volcanic terrain to produce heightened slope instability. Our study shows that the 2023 debris flow event was triggered by an episode of extreme precipitation (Figure 8), which delivered intense rainfall over a short period, rapidly saturating the surface and subsurface soil layers. This event highlights how short-duration storms, increasingly associated with climate change, can overwhelm the buffering capacity of mountainous terrain. The soil media S-7 and S-4, both composed of organic-rich and granular volcanic materials, played a critical role in this response. During the 2023 event, infiltrating rainwater rapidly percolated through these coarse upper layers until reaching the underlying varved glacial sediments (S-2), which have significantly lower permeability. This layering caused a perched water table, increasing the pore pressure and reducing shear strength, ultimately contributing to slope failure. These effects were captured in our remote sensing observations, which showed expanded saturated zones and local instability near the contact between volcanic and glacial deposits.

Our results suggest strong stratigraphic controls and extreme precipitation events in the soils derived from volcanic materials overlying denser glacial layers, acting as failure planes under saturated conditions (Figure 7B; Figure 8B). Our conceptual model promotes water retention and localized pressurization, especially during extreme rainfall events such as 2021 (not observed previously). The conditioning factors are further exacerbated by mid-term climate trends, including the ongoing megadrought in the Southern Andes (Garreaud et al., 2019), which increased desiccation cracking, and weakened root cohesion. Such drought-induced degradation lowers slope resistance, making even moderate precipitation more hazardous. Therefore, our observations suggest that the Ñisoleufu debris flow event, may exemplify the climate-induced changes in both hydrological extremes and landscape memory (drought legacy) act in concert to reduce slope stability. As extreme precipitation becomes more frequent under future climate scenarios, similar failures are

expected to occur across a broader area than observed in past events (Figure 2A). Our findings stress the urgent need for debris flow forecasting models to incorporate stratified soil behaviour, seasonal soil moisture dynamics, and drought-related weakening—factors essential to anticipating the growing hazard posed by climate change (Iverson et al., 2010; Gariano and Guzzetti, 2016).

References:

Garreaud, R. D., Boisier, J. P., Rondanelli, R., Montecinos, A., Sepúlveda, H. H., and Veloso-Aguila, D.: The Central Chile Mega Drought (2010–2018): A climate dynamics perspective, Intl Journal of Climatology, 40, 421–439, https://doi.org/10.1002/joc.6219, 2019.

**Technical corrections:**

1. **TC1_1 L19:** The phrase "influencing by" is unclear. Consider rephrasing as "influenced by" or revising the sentence for clarity.
   - **A:** Thanks, we modified the text.
2. **TC1_2 L20:** Does 'an active area' refer to 'The southern Andes'?
   - **A:** We modify the text to emphasize the area.

     **Original text:**

     This study investigates the generation of the Ñisoleufu debris flow, an active area of debris flow generation, reviewing the interplay between geomorphological, geotechnical and hydrometeorological controls in debris flow dynamics, focusing on the effects of soil properties, slope characteristics and precipitation events.

     **Modified text:**

     This study investigates the generation of the Ñisoleufu debris flow, an active area of debris flow generation in Southern Andes, reviewing the interplay between geomorphological, geotechnical and hydrometeorological controls in debris flow dynamics, focusing on the effects of soil properties, slope characteristics and precipitation events.

3. **TC1_3 L83:** The sentence "Recent events become a significant geological hazard" is unclear.

     **A:** Thanks for this observation, now we have modified the text.

**Original text:**

Recent events become a significant geological hazard especially in steep zones near alluvial plains where human settlements are often established (Fustos et al., 2017; Fustos-Toribio et al., 2021).

**Modified text:**

Recent extreme precipitation events have produced mass wasting hazard, especially in steep zones near alluvial plains where human settlements are often established (Fustos et al., 2017; Fustos-Toribio et al., 2021).

4. **TC1_4** L115: The subheading appears incomplete.

**A:** We agree, now modify the subheading.

**Original text:**

Geomorphological and  geotechnical

**Modified text:**

Geomorphological and  geotechnical conditions

5. **TC1_5 L134-136:** The sentence "Finally, the changes …" is incomplete and ambiguous.

**A:** We agree. Now, we reduced extension and rewrite to be more assertive.

**Original text:**

Finally, the changes in vegetation that occurred as a consequence of the event (Figure 1B), allowing for the inference of the evolution of the post-event landscape and its influence on local eco-geomorphological processes.

**Modified text:**

Finally, we assessed changes in vegetation that occurred as a result of the May 31$^{st}$ event and its subsequent reactivations (Figure 2B), allowing the estimation of the evolution of the landscape.

6. **TC1_6 Table 2:** Please ensure consistency in the formatting of units. For example, "moisture [$w$](%)" and "density[$\rho$](g/cm3)".

**Original text:**

| Soil type/Property | Normative | S-2 | S-3 | S-4 | S-7 |
|---|---|---|---|---|---|
| Moisture [w%] | NCh-1515 | 17.8 | 56.2 | 119.3 | 111.6 |
| Density [ρ] (gr/cm3) | UNE-103-301-94 | 2.07 | 1.52 | <1 | 1.06 |
| Specific Gravity [Gs] | ASTM-D854-14 | 2.76 | 2.49 | 2.5 | 2.34 |
| Liquid Limit [WL] (%) | AS 1289.3.9.1 | 27.48 | 123.93 | - | 149.83 |
| Plastic Limit [WP] (%) | Nch 1517/2 | 16.07 | 91.3 | - | 114.13 |
| Plasticity Index [PL] | NCh1517/2 | 11 | 33 | - | 36 |

| Soil type/Property | Normative | | | | 3.13E-4 |
|---|---|---|---|---|---|
| Hydraulic Conductivity [ku] (m/s) | Porchet and Laferrere (1935) | - | - | - | 3.13E-4 |

**Modified text:**

| Soil type/Property | Normative | S-2 | S-3 | S-4 | S-7 |
|---|---|---|---|---|---|
| Moisture [w] (%) | NCh-1515 | 17.8 | 56.2 | 119.3 | 111.6 |
| Density [$\rho$] (g/cm³) | UNE-103-301-94 | 2.07 | 1.52 | <1 | 1.06 |
| Specific Gravity [$G_s$] | ASTM-D854-14 | 2.76 | 2.49 | 2.5 | 2.34 |
| Liquid Limit [$W_L$] (%) | AS 1289.3.9.1 | 27.48 | 123.93 | - | 149.83 |
| Plastic Limit [$W_P$] (%) | Nch 1517/2 | 16.07 | 91.3 | - | 114.13 |
| Plasticity Index [PL] | NCh1517/2 | 11 | 33 | - | 36 |
| Hydraulic Conductivity [$k_u$] (m/s) | Porchet and Laferrere (1935) | - | - | - | 3.13E-4 |

7. **TC1_7 L439:** 'to' should be removed from 'can to deliver'.

**Original text:**

All the codes used in this manuscript are reproducible from the main text. We can to deliver the main script under any request.

**Modified text:**

All the codes used in this manuscript are reproducible from the main text. We can deliver the main script under any request.

---

## Author Comment (AC2)

**Reviewer #2:**

**General Comment**

**RC2_1** The manuscript titled *"Controls over debris flow initiation in glacio-volcanic environments in the Southern Andes"* presents a comprehensive analysis of the mechanisms responsible for debris flow initiation in the Ñisoleufu sector, Southern Chile. It effectively integrates geomorphological, geotechnical, hydrometeorological, and remote sensing analyses to assess the factors controlling debris flow occurrence in a complex glacio-volcanic setting. The study's use of multi-source data strengthens the reliability of the findings.

However, the manuscript would benefit from a clearer presentation of the connections between controlling factors and observed events, as well as a more explicit discussion of implications for hazard assessment.

> **A:** We appreciate this comment which allowed us to improve both the clarity and scientific value of our manuscript. Following the reviewer's suggestion, we have revised key sections of the manuscript (Abstract, Discussion, Conclusion) to better articulate the connections between the geomorphological, geotechnical, hydrometeorological, and surface deformation factors with the observed debris flow event in Ñisoleufu. In particular, the abstract was slightly modified to reflect this integrated perspective, underscoring the relevance of our findings for improving debris-flow risk forecasting in the Southern Andes. We add the lines 41-42. In the discussion section, we restructured the introductory (lines 335-336) and final (lines 450-452) paragraphs to explicitly outline how each controlling factor contributed to the triggering of the debris flow. This framing now provides a clearer causal linkage between the environmental conditions and the observed event. Finally, we strengthened the conclusions, where we now highlight the broader implications of our findings for hazard assessment and early warning systems in glacio-volcanic terrains. We stress the importance of incorporating hydrogeotechnical and meteorological properties (e.g., tephra-over-varve systems, soil moisture dynamics, extreme rainfalls), into regional debris-flow susceptibility models (lines 478-482).

**Specific Comments**

**Introduction**

**RC2_2** The introduction provides a comprehensive background but could be more focused, particularly in the discussion of previous studies.

**A:** We sincerely appreciate the reviewer's insightful observation. We have added a new paragraph incorporating international case studies to provide a more comprehensive background and strengthen the context of our contribution.

**Original text:**

Debris flows have occurred more frequently due to climate change, resulting in an increase in episodes of extreme rainfall (Jakob and Lambert 2009; Lee 2017; Fustos et al., 2017; Dey and Sengupta 2018) and mainly related to fast changes of soil water content (Fustos et al., 2021). In South America, these common landslide phenomena produce widespread damages, representing a significant threat to human life (e.g. Sepúlveda and Petley 2015; Vega and Hidalgo 2016; Garcia-Delgado et al. 2022). Consequently, the need to forecast (Fustos et al., 2020a) and mitigate (Fustos et al., 2021b) the effects of these events has become a high priority for governments facing increasing episodes of rainfall-induced landslides (RIL) linked with climate change. An accurate forecast needs precise understanding of the triggering conditions of debris flows. South America has limited knowledge of the conditioning factors and triggering conditions of flows, which reduces the capacity of authorities and stakeholders to propose science-based decisions.

Understanding the impact of debris flows in glacial environments is critical in the Chilean southern Andes, particularly due to the most part of the inhabitants lives there. Changes of precipitation patterns related to climate change, particularly fast and intense rainfall events, could amplify the frequency and magnitude of debris flows (Fustos et al., 2022). An increase of extreme hydrometeorological events affecting slopes in glacial settings is observed, whose mechanical properties and geomorphology have evolved since the Last Glacial Maximum (Fustos-Toribio et al., 2021; Somos-Valenzuela et al., 2020; Ochoa-Cornejo et al. 2025). Despite the significance of these events to the population, considerable uncertainty remains about how the interaction between volcanic-derived soils over glaciar landforms will respond to extreme hydrometeorological events.

The landslide generation from extensional failures to surface deformation stands out, mainly due to gravity and surface erosion in high precipitation environments (Xie et al., 2020; Yi et al., 2021). Over time, these failures expand and deepen, weakening the soil and predisposing it to landslides, especially under water-saturated conditions or heavy rainfall (Fustos et al., 2017; Wang et al., 2024). The capacity to oversee these extensional failures in remote areas close to roads is an open question yet, mainly in South America. Large landslides can, under certain conditions, transform into debris flows when the slid material mixes with water, increasing its fluidity like the Villa Santa Lucia event in the Chilean Patagonia (Somos-Valenzuela et al., 2020). These debris flows are fast and destructive, capable of

transporting large amounts of material and causing significant damage to infrastructure and ecosystems, as well as posing a serious hazard to nearby communities (Hirschberg et al., 2021). The occurrence of debris flows in volcanic environments is a subject of significant scientific interest (Cheung & Giardino, 2023), primarily due to the intricate nature of volcanic soils and their inherent textural variability (Thompson et al., 2023), which directly impacts water content dynamics. In the Southern Andes region, there remains a conspicuous lack of understanding regarding how these textural variations influence the hydraulic response of these soils during extreme hydrometeorological events. Over the past four decades, this area has witnessed substantial volcanic activity (Moreno-Yaeger et al., 2024), resulting in extensive deposition of tephra that has significantly contributed to the heightened occurrence of debris slides and debris flows (Korup et al., 2019). Despite these recurrent occurrences, substantial gaps persist in comprehending the variability of textural and hydraulic properties of volcanic soils under the influence of extreme hydrometeorological events. Previous studies suggested the need to seek and address the textural and hydraulic properties of volcanic soils, constraining the hydraulical and geomechanical conditions that enable the debris flow generation (Fustos et al., 2021; Kostynick et al., 2022). These constraints become essential to generate accurate planning over the territory, allowing to save lives and develop accurate early warning systems.

[revised manuscript text omitted]

**RC2_3** Some references to regional climate impacts could be condensed to streamline the narrative.

**A:** We agree. Now, we introduced additional information in the first part of a new second paragraph.

**Additional information in second paragraph:**

Worldwide, the increasing frequency and intensity of such extreme precipitation events exacerbated by climate change to regional scale (Stoffel et al., 2013; Pavlova et al., 2014) introduce severe threats to human life and property. Changing precipitation patterns related to extreme precipitation, lead to conditions conducive for debris flows in wide areas in North America (Bovis et al., 1999), Asia (Ma et al., 2013; Chang et al., 2017), Europe (Malet et al., 2005; Stoffel et al., 2013; Pavlova et al., 2014) and South America (Sepúlveda et al., 2013; Sepúlveda et al., 2014; Fustos et al., 2022). Stand out heavy tropical storms, such as Taiwan, increasing debris flow incidents (Chang et al., 2024) and compromising the safety of urban areas located near mountainous terrains (Chen et al., 2015; , Kang et al., 2017). The Wenchuan Earthquake in China exemplifies how seismic activity can trigger extensive debris flows, resulting in not only immediate destruction but also long-lasting hazards due to the formation of landslide-dammed lakes that threaten downstream communities (Cui et al., 2009; Ma et al., 2013; Wang & Yan, 2015). Accurate forecasting of debris flow events is vital for disaster preparedness and mitigation. Understanding the triggering conditions— including rainfall intensity, groundwater levels, and geological features—is

essential for developing conceptual models that represent the regional conditions that could control a debris flow initiation. Debris flows are influenced by soil hydraulic characteristics and the intensity/duration of rainfall events (Singh and Kumar, 2021), in which rainfall intensities serve as crucial predictors in mountainous regions (Chang et al., 2017; Fustos-Toribio et al., 2022). Moreover, coarse-grained volcanic soils exhibit transient increases in pore pressure during intense rainfall events (Huang et al., 2012). Conversely, fine-grained soils with low infiltration rates do not experience significant changes in the pore pressure, generating failures due to decreased soil shear strength (Dahal et al., 2008; 2011). Hence, understanding soil composition and granulometric features is pivotal in assessing debris flow susceptibility worldwide. Debris flows mainly controlled by the geomorphological features and the specific geological evolution of each region of the planet, highlighting the need for localized and context-specific approaches for their study and management. One of the next frontier correspond to constraint the debris flow generation under glacial environment under changing precipitation events related to climate change.

**RC2_3** The citations provided lack robustness for a proper framing of debris flows, as most references are from the last 3-4 years. Correct this by adding relevant and foundational studies.

> **A:** We agree and appreciate this observation. We have introduced a deep review of events and literature review, considering the poor understanding of debris flow events in South America. We focused on the Wos contribution with high influence in the field, considering citing metrics.

> **Literature now included:**

> Bovis, M. J. and Jakob, M.: The role of debris supply conditions in predicting debris flow activity, Earth Surf. Process. Landforms, 24, 1039–1054, https://doi.org/10.1002/(sici)1096-9837(199910)24:11<1039::aid-esp29>3.0.co;2-u, 1999.

> Gregoretti, C.: The initiation of debris flow at high slopes: experimental results, Journal of Hydraulic Research, 38, 83–88, https://doi.org/10.1080/00221680009498343, 2000.

> Ma, C., Hu, K., and Tian, M.: Comparison of debris-flow volume and activity under different formation conditions, Nat Hazards, 67, 261–273, https://doi.org/10.1007/s11069-013-0557-6, 2013.

> Malet, J.-P., Laigle, D., Remaître, A., and Maquaire, O.: Triggering conditions and mobility of debris flows associated to complex earthflows, Geomorphology, 66, 215–235, https://doi.org/10.1016/j.geomorph.2004.09.014, 2005.

Pavlova, I., Jomelli, V., Brunstein, D., Grancher, D., Martin, E., and Déqué, M.: Debris flow activity related to recent climate conditions in the French Alps: A regional investigation, Geomorphology, 219, 248–259, https://doi.org/10.1016/j.geomorph.2014.04.025, 2014.

Stoffel, M., Mendlik, T., Schneuwly-Bollschweiler, M., and Gobiet, A.: Possible impacts of climate change on debris-flow activity in the Swiss Alps, Climatic Change, 122, 141–155, https://doi.org/10.1007/s10584-013-0993-z, 2013.

**RC2_4** Additionally, the transition between topics is somewhat abrupt, lacking fluidity.

**A:** We sincerely appreciate the reviewer's comments and observations and apologise for the shortcomings in the original writing, which did not convey the intended focus or depth of analysis. We have carefully revised and restructured the text to enhance its coherence and scientific rigour.

In the revised version, the first paragraph now provides a more focused description of the core issue, emphasising the transformation of landslides into debris flows in volcanic settings under intense rainfall, and highlighting the critical role of the textural variability of volcanic soils. The second paragraph expands the discussion to a global perspective, incorporating recent and foundational literature to address comment **RC2_3**. This broader context helps situate our case study within a widely recognised scientific problem. The third paragraph underscores the need to better understand and constrain the controlling factors in glacio-volcanic environments, where geomorphological and hydrometeorological extremes can significantly influence debris flow initiation. This section shows the importance of integrating local characteristics into such analyses. The fourth paragraph narrows the focus to the Southern Andes, presenting two iconic case studies—Osorno and Villa Santa Lucía. These examples illustrate existing limitations in our understanding of conditioning factors, particularly regarding water content and the properties of mobilised material. Finally, the fifth paragraph concludes the section by emphasising the urgency of developing integrated conceptual models to support effective land-use planning.

**Methodology**

It would be important to include the location of the meteorological stations in one of the displayed figures to provide spatial context for the data collection.

**A:** We agree, now we include an additional figure as Figure 1, includng the location of the meteorological stations.

**Figure 1** Rainfall-Induced mass wasting in Southern Andes. A) Regional map of Ice-sheet extension during 35 ka and 20 ka as an example, and volcanoes emplaced in the area. B) Zoom to study area with Ñisoleufu in Northern Ice sheet sector showing the

weather stations. C) Zoom to Northern Patagonian area showing correlation between mass wasting events and moraine lines (blue line). D) Zoom to the Osorno volcano area showing the high debris flow generation area discussed in Fustos et al., 2022.

[Figure]

**Results**

The results are comprehensive, but the presentation is occasionally repetitive. For example, the discussion of soil stratigraphy (Lines 200-240) could be condensed to highlight the most critical observations.

**A:** We appreciate the reviewer's constructive suggestion. We agree that the section describing soil stratigraphy can be condensed to emphasize the key features relevant to the mechanisms of soil saturation and landslide triggering. Accordingly, we have revised Lines 200–250 to focus on the most significant stratigraphic observations, namely: the alternation between volcanic ash-falls and organic soil development over a low-permeability glacial substrate. This reorganization improves the clarity of the results and avoids unnecessary repetition.

**Original text in lines (200-240).**

In relation to the channelled flow deposits, the blocks and gravels are predominantly (~75%) composed of material from Sierra Quinchilca, while a smaller proportion (~25%) is composed of granodiorite from the Futrono-Riñihue Batholith (Figure 4D). The composition of the matrix deposit is derived from the slope volcanic and glacial deposits (Rodríguez et al., 1999). This geological and soil composition is representative of a typical distribution in the Southern Andes, specifically spanning the latitudinal range from 39° to 42°S.

Below the steepest slope zone of the hillslope, sequences with significant spatial variability were identified (Figure 4B-D). At 350 m a.s.l. (column C3) slopes vary

between 30 up to 45°. The basement associated with the CPfgr crops out at the base, in erosive contact with level S-1, mainly related to glacial deposits with the presence of large angular blocks. Laminar levels of clay and silt towards the top (S-2) stand out. Above the glacial deposit is level S-3. At first glance, the paleosol observed in c2 appears to have a relative age younger than the glacial deposits but older than the Neltume ashfall deposit. However, field observations indicate that paleosol S-3 is younger than the Neltume ashfall, as level S-4 in c3 represents a reworked deposit derived from this ashfall event (S-4). The presence of imbricated pyroclasts and their noticeable variation in thickness (Figure 3D) support this interpretation. Finally, at the top of the sequence illustrated in columns c3 and c4, level S-7 shows active soil development, reaching approximately half of the thickness observed in column c1. Near the base of the flow, at an altitude of 250 m a.s.l. and with slopes between 0 and 20°, a sequence without the levels observed in the previous points upstream was identified (Figure 4). The area represented by column C4, showed soil levels associated with previous mass wasting events (Figure 3E). At the base, level S-5 is identified, associated with a polymictic deposit with a high presence of clasts, reaching a size up to 10 cm, and slight northern imbrication. The matrix of the deposit has pumice content from the Neltume deposit (S-4), related to the explosive event of the Mocho-Choshuenco volcanic complex, and clays from the glacial deposits (S-1 and S-2), present upstream as observed in columns c1 and c3. Above the flow is another level (S-6) with the same characteristics, but with a lower proportion of clasts with respect to the matrix. Finally, towards the top, the development of the current soil (S-7) is observed, with a thickness like that measured in column c3.

**Modified text in lines (200-240).**

Stratigraphic analysis along the slope (Figure 5B–D) reveals a consistent sequence overlying the CPfgr basement, characterized by a basal glacial deposit (S-1) with large angular clasts, overlain by silt–clay layers (S-2). These low-permeability units are covered by interbedded paleosols and volcanic ash-falls. In column C3, the paleosol S-3 was initially considered older than the Neltume ashfall, but the presence of reworked pyroclasts in S-4 (Figure 4D) suggests a younger relative age. The upper sequence concludes with active soil formation (S-7; Figure 5D).

Further downslope (250 m a.s.l.), column C4 exhibits a distinct stratigraphy dominated by mass-wasting deposits. The basal level (S-5) consists of a polymictic unit with sub-rounded clasts and pumice fragments from the Neltume event, embedded in a clay-rich matrix derived from upstream glacial units. A similar, though finer, deposit (S-6) overlies S-5, and is capped by the same modern soil unit (S-7). This stratigraphic framework—composed of alternating low-permeability substrates and reworked tephras—is representative of Andean terrains between 39° and 42°S and is critical to understanding hydrological storage and slope instability (Figure 1).

Figure 6A does not appear to indicate an anomalous trend in soil moisture compared to previous years. It should be emphasized that the combination of saturated soil and intense rainfall plays a key role in triggering debris flows.

**A:** Thanks for this comment. We agree the point out of the reviewer. Now, we emphatized the idea in the section 4.2 considering a rewriting of a paragraph and an additional discussion (and limitations of the study) in sections 5.2 and 5.3

**Original text in section 4.2:**

Analysis using the ERA5 model reveals a pronounced annual cycle in soil moisture (Figure 6). Antecedent precipitation data indicated a surface saturation, showing a shallow condition reaching a full saturation in the first metre of depth, mainly related to previous rainfall events (Figure 6B-C). Moreover, a substantial increase in soil moisture was identified at greater depths (up to 2.89 meters), exhibiting a rapid 50% change preceding the onset of flow events (Figure 6B). This notable fluctuation in soil moisture occurred specifically at the interface between tephra and till-varves, implying potential implications for the natural stability of the terrain base.

**Modified text in section 4.2:**

Analysis using the ERA5 model reveals a well-defined annual cycle in soil moisture (Figure 8A). While the overall evolution of moisture content does not indicate an anomalous trend when compared to previous years, significant short-term variations in soil moisture were observed (Figure 8B) in response to extreme precipitation events (Figure 8C). The preliminary results showed that antecedent rainfall led to surface saturation, with moisture levels reaching full saturation within the first metre of soil depth, primarily due to accumulated rainfall in the days prior (Figure 8B–C). Notably, a substantial and rapid increase in soil moisture was also detected at greater depths—up to 2.89 meters—where a 50% change occurred shortly before the onset of debris flow events (Figure 8B). This marked fluctuation was concentrated at the interface between tephra and till-varves, suggesting critical implications for slope instability. Our findings underscore that, even in the absence of a long-term anomalous trend, the combination of saturated soil conditions and intense rainfall plays a decisive role in triggering debris flows.

**Added paragraph to section 5.2:**

Climate change is increasingly debris flow generation by altering precipitation patterns and soil moisture dynamics (Talebi et al., 2007). In the Southern Andes, volcanic soils with variable textures play a critical role in this process. Enhanced seasonal moisture variability, exacerbated by extreme precipitation, leads to fast soil saturation, especially where fine-grained soils form low-permeability layers above coarser materials (Figure

10). These stratified soil conditions promote subsurface water storage, increasing the slope instability under saturated conditions (Talebi et al., 2007). Fine volcanic over glacial deposits can act as lubricants, further weakening slope cohesion and promoting failure (Espinosa et al., 2024) during intense rainfall, as happened during the Ñisoleufu event (Figure 7B). This event highlights how short-duration storms, increasingly associated with climate change, can overwhelm the buffering capacity of mountainous terrain. The soil media S-7 and S-4, both composed of organic-rich and granular volcanic materials, played a critical role in this response. During the 2023 event, infiltrating rainwater rapidly percolated through these coarse upper layers until reaching the underlying varved glacial sediments (S-2), which have significantly lower permeability. This layering caused a perched water table, increasing the pore pressure and reducing shear strength, ultimately contributing to slope failure. These effects were captured in our remote sensing observations, which showed expanded saturated zones and local instability near the contact between volcanic and glacial deposits.

**Discussion**

**C:** The discussion appropriately addresses the key findings but could benefit from a more critical evaluation of the limitations of the study.

> **A:** We agree. Now, we added additional information about the limitation of PS data as limitation. Moreover, we discuss in detail the main limitations in section 5.2 and 5.3.
>
> New paragraph in 5.2 related to limited PS
>
> > Our results showed a limited amount of PS in the study area similar to previous studies in the area (Vasquez-Antipan et al., 2025). The Southern Andes, and the Ñisoleufu area, is characterized by complex geomorphological features and varying precipitation patterns that could introduce uncertainty to the remote sensing measurements. The application of limited persistent scatterer data in assessing slope deformations offers a promising avenue for the development of a landslide early warning system (LEWS) in the Southern Andes. However, the investigation into this method requires a thorough understanding of potential limitations, particularly the extensive vegetation cover in the Southern Andes, which can obscure satellite signals and affect data accuracy. Vegetation serves as a significant barrier to radar signals, leading to incomplete datasets that might obscure important geological signals indicative of slope movements (Maragaño-Carmona et al., 2023). Therefore, additional efforts must be considered to move forward to an operational scale.

New paragraph in 5.3

Our study contributes to understanding the relationship between volcanic and glacial deposits under extreme rainfall events forcing in the Southern Andes. However, further research is necessary to improve the RIL susceptibility models due to an incomplete integration of critical soil properties. Several studies focus on rainfall thresholds as primary trigger, oversimplifying the failure mechanisms. Our study proposes that additional assessment of the hydraulic and mechanical influence of specific soil layers must be considered. Additionally, the pronounced spatial heterogeneity in soil layer composition in Southern Andes, ranging from S-1 to S-7, and variability in thickness and permeability, further complicates predictive accuracy to regional scale. Future developments must consider high-resolution subsurface mapping introducing national scale models (Dinamarca et al., 2024), allowing better RIL risk models overlooking zones where possible saturated soil could appear, leading to sudden failure. Moreover, surface deformation frequently observed as a slow extensional signal prior to collapse introduces a limited amount of PS. Moreover, the hydrometeorological variability indicates that better soil moisture models are necessary in the zone to improve the slope stability analysis. Together, these limitations highlight the urgent need for multidisciplinary approaches that integrate geotechnical, geomorphological, and hydrometeorological data into landslide hazard assessments.

**C:** The role of climate change in debris flow initiation is briefly mentioned but not fully explored. Given the relevance of this factor, a more detailed discussion would enhance the manuscript's impact.

**A:** We agree. We have now included a more in-depth discussion of our study and its implications, taking into account the new Figure 1 and the scientific results. This discussion has been incorporated into Sections 5.2 and 5.3.

**New Figure:**

**Figure 1** Rainfall-Induced mass wasting in Southern Andes. A) Regional map of Ice-sheet extension during 35 ka and 20 ka as an example, and volcanoes emplaced in the area. B) Zoom to the study area with Ñisoleufu in the Northern Ice Sheet sector, showing the weather stations. C) Zoom to Northern Patagonian area showing correlation between mass wasting events and moraine lines (blue line). D) Zoom to the Osorno volcano area showing the debris flow release area discussed in Fustos et al., 2022.

[Figure]

**New paragraph in section 5.2**

Climate change is increasingly debris flow generation by altering precipitation patterns and soil moisture dynamics (Talebi et al., 2007). In the Southern Andes, volcanic soils with variable textures play a critical role in this process. Enhanced seasonal moisture variability, exacerbated by extreme precipitation, leads to fast soil saturation, especially where fine-grained soils form low-permeability layers above coarser materials (Figure 10). These stratified soil conditions promote subsurface water storage, increasing the slope instability under saturated conditions (Talebi et al., 2007). Fine volcanic over glacial deposits can act as lubricants, further weakening slope cohesion and promoting failure (Espinosa et al., 2024) during intense rainfall, as happened during the Ñisoleufu event (Figure 7B). This event highlights how short-duration storms, increasingly associated with climate change, can overwhelm the buffering capacity of mountainous terrain. The soil media S-7 and S-4, both composed of organic-rich and granular volcanic materials, played a critical role in this response. During the 2023 event, infiltrating rainwater rapidly percolated through these coarse upper layers until reaching the underlying varved glacial sediments (S-2), which have significantly lower permeability. This layering caused a perched water table, increasing the pore pressure and reducing shear strength, ultimately contributing to slope failure. These effects were captured in our remote sensing observations, which showed expanded saturated zones and local instability near the contact between volcanic and glacial deposits.

**New text introduced in section 5.3**

On a regional scale, climate change is intensifying debris flow hazards worldwide (Gariano and Guzzetti, 2016). In the Southern Andes, current changes in precipitation patterns will affect the stability of volcanic and glacial deposits through alterations in water storage, as noted in this study (Figure 8B). The region's unique stratigraphy in South America, where volcanic soils overlay glacial sediments, may become unstable during extreme rainfall events. Significant shifts in precipitation patterns, as predicted by CMIP6 models, alter the spatial distribution of precipitation and their impact on soil moisture storage, with limited accurate estimations (Salazar et al., 2023). We propose that future developments should carefully constrain areas with high susceptibility to debris flow. Therefore, improved hazard debris flow delimitation and instrumental monitoring become critical for reducing the impact of these hazards in the Southern Andes.

Our study contributes to understanding the relationship between volcanic and glacial deposits under extreme rainfall events forcing in the Southern Andes. However, further research is necessary to improve the RIL susceptibility models due to an incomplete integration of critical soil properties. Several studies focus on rainfall thresholds as primary trigger, oversimplifying the failure mechanisms. Our study proposes that additional assessment of the hydraulic and mechanical influence of specific soil layers must be considered. Additionally, the pronounced spatial heterogeneity in soil layer composition in Southern Andes, ranging from S-1 to S-7, and variability in thickness and permeability, further complicates predictive accuracy to regional scale. Future developments must consider high-resolution subsurface mapping introducing national scale models (Dinamarca et al., 2024), allowing better RIL risk models overlooking zones where possible saturated soil could appear, leading to sudden failure. Moreover, surface deformation frequently observed as a slow extensional signal prior to collapse introduces a limited amount of PS. Moreover, the hydrometeorological variability indicates that better soil moisture models are necessary in the zone to improve the slope stability analysis. Together, these limitations highlight the urgent need for multidisciplinary approaches that integrate geotechnical, geomorphological, and hydrometeorological data into landslide hazard assessments.

**Conclusions**

The conclusions effectively summarize the main findings but lack specific recommendations for hazard mitigation. Consider adding a few practical insights based on the study's results.

**A:** Thanks for this comment. We agree with the point of the reviewer. Now, we emphasize this idea in the last paragraph of Conclusion.

**New paragraph in Conclusions:**

Finally, our results demonstrate the value of integrating geomorphological, hydrometeorological, and hydrogeotechnical data to support debris-flow hazard assessments. In particular, the combination of tephra layers overlying low-permeability glacial deposits, together with rapid water infiltration on steep slopes during extreme rainfall events, defines a critical configuration that enhances susceptibility to failure. This integrated framework provides a robust basis for identifying high-risk areas and strengthening early warning strategies in the Southern Andes and comparable volcanic-glacial settings worldwide.

---

## Author Comment (AC3)

**Additional specific Comments**

Upon further consideration I have identified additional points that may require other clarifications:

1. Laboratory Measurements (Table 2): The values reported for liquid limit (WL) and plastic limit (WP) for S-3, as well as WL, WP, and Ku for S-7, appear inconsistent with their geological descriptions. S-3. S-7 represents an active, developing soil, likely rich in organic material, where higher hydraulic conductivity (Ku) would be expected. However, the reported Ku is surprisingly low, potentially indicating compaction, fine-grained content, or measurement errors. Please verify these values and, if accurately, provide additional context to justify these unexpected results.

   **A:** We verify the results. Our results show similar range of values in comparison to Vasquez-Antipan et al., (2025) and volcanic soils in Southern Andes. Now, we introduced additional information in discussion (section 5.2)

   **Original text in section 5.2:**

   Moreover, field evidence suggests that the Ñisoleufu event is not an isolated case as seen in the remobilised events (Figure 1, geological map - alluvial deposit: Ha). The geotechnical properties of the material to be remobilised are crucial for establishing stability conditions. The granulometric characteristics of the deposits, primarily granular types associated with S-4, are identified as frictional soils overlying fine-grained, cohesive soils like varves (S-2). Other soils in Southern Andes, such as S-3 and S-7, could be originated from the decomposition of volcanic glass from ashes and glacial clays (Sanhueza et al., 2011), resulting in particles smaller than 0.1 mm (Figure 5). The distribution of the soil layers varies abruptly downslope, as observed in columns c1 and c3 for S-1, S-2, and S-4, indicating intense mass wasting and erosion productivity in areas close to glacial lakes (Figure 4D). The frictional soils, related to S-4, exhibit high shear resistance (Chen et al, 2021), combined with steep slopes, can contribute to stability control of post-glacial volcanic deposits (Walding et al, 2023; Ontiveros-Ortega et al, 2023). However, while frictional soils are generally more resistant to sliding (Chen et al, 2021), soil saturation can significantly decrease their strength, thus increasing the risk of failure under extreme precipitation events detected in recent years in the Southern Andes (Fustos et al., 2017; Somos-Valenzuela et al., 2020; Fustos et al., 2021). This is consistent with the presence of extensional failure observed before flow initiation and subsequent reactivations in June 2023 and 2024 (Figure 3C; Figure 8).

   **Modified text in section 5.2:**

Moreover, field evidence suggests that the Ñisoleufu event is not an isolated case, as indicated by other remobilised events in the area (Figure 2, geological map – alluvial deposit: Ha). The geotechnical properties of the remobilised materials are critical for defining slope stability conditions. Granulometric analyses indicate that the deposits are primarily granular soils, such as those associated with S-4, which are classified as frictional and are found overlying finer-grained cohesive soils, such as varves (S-2). Other soils found in the Southern Andes, including S-3 and S-7, originate from the decomposition of volcanic glass and glacial clays (Sanhueza et al., 2011; Vasquez et al., 2025), producing particles smaller than 0.1 mm (Figure 5).

Specifically, S-3 soils, derived from explosive eruptions of the Mocho-Choshuenco volcano, consist of non-cohesive volcanic ash mixed with fine-grained sediments, forming a matrix with elevated plasticity and a high liquid limit (Vasquez et al., 2025). These properties result from the introduction of fine material during the deposition. Moreover, S-7 soils, classified as organic soils derived from volcanic deposits, exhibit notably high liquid limits due to the accumulation of organic matter. The organic matter enhances the soil's water retention and promotes the formation of organic colloids, which may increase the liquid limit (Deng et al., 2017; Fiantis et al., 2019). Our results are consistent with independent laboratory testing in the zone (Vásquez et al., 2025), which shows that organic-rich paleosols were buried after the Last Glacial Maximum, approximately 5 km south of the study area, and exhibit similar liquid limit values to those observed in S-7.

The spatial distribution of soil layers varies abruptly along the slope, as observed in columns C1 and C3 for S-1, S-2, and S-4, indicating significant mass wasting and erosion processes near glacial lakes (Figure 5D). The frictional soils, such as those related to S-4, generally exhibit high shear strength (Chen et al., 2021), and when combined with steep topography, may contribute to the relative stability of post-glacial volcanic deposits (Walding et al., 2023; Ontiveros-Ortega et al., 2023). However, under extreme precipitation events—such as those recorded in recent years in the Southern Andes—soil saturation can substantially reduce the strength of even frictional soils, increasing the likelihood of failure (Fustos et al., 2017; Somos-Valenzuela et al., 2020; Fustos et al., 2021). This mechanism aligns with the observed extensional failures that preceded the initiation and reactivation of flows in June 2023 and 2024 (Figure 4C; Figure 10).

○ **References:**

Deng, Y., Cai, C., Xia, D., Shuwen, D., Chen, J., & Wang, T. (2017). Soil atterberg limits of different weathering profiles of the collapsing gullies in the hilly granitic region of southern china. Solid Earth, 8(2), 499-513. https://doi.org/10.5194/se-8-499-2017

Fiantis, D., Ginting, F., Gusnidar, G., Nelson, M., & Minasny, B. (2019). Volcanic ash, insecurity for the people but securing fertile soil for the future. Sustainability, 11(11), 3072. https://doi.org/10.3390/su11113072

Ustiatik, R., Ariska, A., Hakim, Q., Wicaksono, K., & Utami, S. (2023). Volcanic deposits thickness and distance from mt semeru crater strongly affected phosphate solubilizing bacteria population and soil organic carbon. Journal of Ecological Engineering, 24(10), 360-368. https://doi.org/10.12911/22998993/170860

2. Limited Number of PS (Figure 7): The analysis includes only five PS, of which three are located near the boundary and only two within the landslide niche. This sparse dataset may not provide a sufficiently robust basis for reliable precursor identification, especially given that these PS do not appear to exhibit significantly different displacement patterns compared to surrounding points. This raises questions about

the reliability of these PS as early warning indicators. I recommend adjusting the color scale in Figure 7 to better distinguish between PS displacement and elevation, which may improve the interpretability of the figure.

A: We thank the reviewer for the valuable observation regarding the limited number and spatial distribution of Persistent Scatterers (PS) included in our analysis. We acknowledge that the sparse dataset, comprising a reduced number of PS, limits the robustness of our results. As noted by the reviewer, the displacement patterns of these PS do not markedly differ from surrounding points, which indeed constrains their immediate utility as early warning indicators. We consider that this is a preliminary study, and we agree that surface deformation data alone are insufficient for the development of a reliable early warning system. Further investigation is needed, particularly to assess local conditions such as dense vegetation cover, which poses additional challenges for PS detection in the southern Andes.

Nonetheless, previous studies in the area have demonstrated the feasibility of using PS data as complementary information to support landslide hazard assessments (Vasquez-Antipan et al., 2025). In our case, although the InSAR-derived displacements are not conclusive on their own, they are consistent with geomorphological indicators observed during fieldwork, which suggest evidence of previous surface deformation in the area. These converging lines of evidence support the hypothesis of progressive slope instability, despite the limitations of the remote sensing dataset. Now, we discuss in detail it in the section 5.2 and 5.3 as limitation of the study and future scope.

**Introduced paragraph in section 5.2:**

> Our results showed a limited amount of PS in the study area similar to previous studies in the area (Vasquez-Antipan et al., 2025). The Southern Andes, and the Ñisoleufu area, is characterized by complex geomorphological features and varying precipitation patterns that could introduce uncertainty to hte remote sensing measurements. The application of limited persistent scatterer data in assessing slope deformations offers a promising avenue for the development of a landslide early warning system (LEWS) in the Southern Andes. However, the investigation into this method requires a thorough understanding of potential limitations, particularly the extensive vegetation in the Southern Andes, which can obscure satellite signals and affect data accuracy. Vegetation serves as a significant barrier to radar signals, leading to incomplete datasets that might obscure important geological signals indicative of slope movements (Maragaño-Carmona et al., 2023). Therefore, additional efforts must be considered to move forward to an operational scale.

> In response to the reviewer's suggestion, we will revise Figure 7 by adjusting the color scale to enhance the visual distinction between PS displacement and elevation. We believe this improvement will increase the figure's interpretability and help contextualize the deformation patterns within the topographic setting.

Original Figure:

[Figure]

Modified Figure:

[Figure]

**Reference:**

Vásquez-Antipán, D., Fustos-Toribio, I., Riffo-López, J., Cortez-Díaz, A., Bravo, Á., and Moreno-Yaeger, P.: Landslide processes related to recurrent explosive eruptions in the Southern Andes of Chile (39° S), Journal of South American Earth Sciences, 157, 105469, https://doi.org/10.1016/j.jsames.2025.105469, 2025.

3. To improve clarity, consider presenting the data in Table 3 as a graph, which may provide a more intuitive visualization of the variation in saturated hydraulic conductivity across different soil types.

   **A:** We agree. Now, we modify the table 3

   **Original form:**

**Table 3 Hydraulic properties of release zone in debris flow generation zone (column c1 in Figure 1).**

| Layer | depth (m) | Ks @10C (m/s) | Description |
|---|---|---|---|
| Superior layer | 0-0.5 | 3.31E-04 | Organic (S-7) |
| Volcanic deposit 1 | 0.5-2.5 | 2.24E-04 | Neltume ashfall deposit (S-4) |
| Volcanic deposit 2 | 2.5-2.7 | 4.64E-05 | base Neltume (S-4) |
| Varve | 2.7-3.0 | 1.54E-05 | Varves (S-2) |
| Morraine | 3.0-?? | 2.65E-05 | Saturated Morraine (S-1) |

**Version in Figure:**

[Figure]

**Additional modifications**

We modified the granulometric figure (Figure 6).

Original figure:

[Figure]

Modified figure:

---

## Author Response (AR2)

**Controls over debris flow initiation in glacio-volcanic environments in the Southern Andes**

Response of authors

We thank you very much for the detailed and thoughtful review. We truly appreciate the time and care you dedicated to carefully assessing the manuscript, considering these dates. We appreciate the positive feedback. Following, we indicate the modifications introduced to the document in this second round.

The authors have conducted a comprehensive revision and provided careful responses to all the comments. The manuscript is now better structured, and the representation of the hazard model, where volcanic material overlies glacial deposits has been improved by the inclusion of figure 1. Nevertheless, some technical revisions are still needed.

>    A: Thank you for the detailed review of the document.

1. L50: It is unnecessary to abbreviate 'rainfall-induced landslides', as the term is mentioned only a few times throughout the manuscript and is not lengthy.

>    A: Thanks for your comment. We have now removed the abbreviation from the abstract. Moreover, the manuscript was modified from RIL to mass wasting in the context that these events are triggered by rainfall.

2. L444: The term 'localized pressurization' introduced in the newly added paragraph has not been discussed previously. Could the authors please elaborate on this in more detail?

A: Our apologies for the modification. We now continue the sequence of the document, changing from "soil pore pressure" used in the introduction and study zone sections.

Original text:

> Our conceptual model promotes water retention and localized pressurization, especially during extreme rainfall events such as 2021

Modified text:

> Our conceptual model promotes water retention and changes in soil pore pressure, especially during extreme rainfall events such as 2021

3. The authors mention that the Osorno volcano area frequently experiences debris flows and refer to the Petrohue event as an analogue to the Nisoleufu debris flow. It would be interesting, if possible, to include all such analogues in figure 1D.

A: The analogue situation are related to the climatic features and conditioning factors related to volcanic environment. Figure 1 illustrates a spatial description of mass wasting in the southern Andes and its correlation with the Last Glacial Maximum. We think that the current form could introduce bias and misunderstanding of the information provided. Therefore, we rewrite the sentence of the discussion in 5.3

Original text:

> The Ñisoleufu debris flow showed a characteristic pattern of mass wasting processes in the Southern Andes, becoming analogues to Petrohue event (Fustos-et al., 2021) in Osorno Volcano (Figure 1D).

Modified text:

> The Ñisoleufu debris flow exhibited a characteristic pattern of mass-wasting processes in the Southern Andes, similar to the Petrohué event (Fustos et al., 2021) at Osorno Volcano (Figure 1D), which can be attributed to comparable climatic and volcanic conditions.

4. L493: Please remove the word 'worldwide' from the sentence beginning with 'on a regional scale'.

A: Now, we removed the word "worldwide" in line 500.

Additional comment:

Mr Ebel: Your Table 2 contains coloured cells. Please note that this will not be possible in the final revised version of the paper due to HTML conversion of the paper. When revising the final version, you can use footnotes or italic/bold font. For now, the process will continue, but please note that the final version cannot be published by using coloured tables.

A: Now, we modified Table 2

Original table:

| Soil type/Property | Normative | S-2 | S-3 | S-4 | S-7 |
|---|---|---|---|---|---|
| Moisture [w] (%) | NCh-1515 | 17.8 | 56.2 | 119.3 | 111.6 |
| Density [ρ] (g/cm³) | UNE-103-301-94 | 2.07 | 1.52 | <1 | 1.06 |
| Specific Gravity [$G_s$] | ASTM-D854-14 | 2.76 | 2.49 | 2.5 | 2.34 |
| Liquid Limit [$W_L$] (%) | AS 1289.3.9.1 | 27.48 | 123.93 | - | 149.83 |
| Plastic Limit [$W_P$] (%) | Nch 1517/2 | 16.07 | 91.3 | - | 114.13 |
| Plasticity Index [PL] | NCh1517/2 | 11 | 33 | - | 36 |
| Hydraulic Conductivity [$k_u$] (m/s) | Porchet and Laferrere (1935) | - | - | - | 3.13E-4 |

Modified table:

| Soil type/Property | Normative | S-2 | S-3 | S-4 | S-7 |
|---|---|---|---|---|---|
| Moisture [w] (%) | NCh-1515 | 17.8 | 56.2 | 119.3 | 111.6 |
| Density [ρ] (g/cm³) | UNE-103-301-94 | 2.07 | 1.52 | <1 | 1.06 |
| Specific Gravity [$G_s$] | ASTM-D854-14 | 2.76 | 2.49 | 2.5 | 2.34 |
| Liquid Limit [$W_L$] (%) | AS 1289.3.9.1 | 27.48 | 123.93 | - | 149.83 |
| Plastic Limit [$W_P$] (%) | Nch 1517/2 | 16.07 | 91.3 | - | 114.13 |
| Plasticity Index [PL] | NCh1517/2 | 11 | 33 | - | 36 |
| Hydraulic Conductivity [$k_u$] (m/s) | Porchet and Laferrere (1935) | - | - | - | 3.13E-4 |